

# Developing an integrated assessment model to explore optimal cost-benefit paths for Shared Socioeconomic Pathways scenarios

Xuanming Su[1,2], Kiyoshi Takahashi[1], Tokuta Yokohata[3], Katsumasa Tanaka[3,4], Shinichiro Fujimori[5], Jun'ya Takakura[1], Rintaro Yamaguchi[1], and Weiwei Xiong[4]

[1]Social Systems Division, National Institute for Environmental Studies (NIES), Tsukuba, Japan
[2]Research Institute for Global Change/Research Center for Environmental Modelling and Application, Japan Agency for Marine-Earth Science and Technology (JAMSTEC), Yokohama, Japan
[3]Earth System Division, National Institute for Environmental Studies (NIES), Tsukuba, Japan
[4]Laboratoire des Sciences du Climat et de l'Environnement (LSCE), IPSL, CEA/CNRS/UVSQ, Université Paris-Saclay, Gif-sur-Yvette, France
[5]Department of environmental engineering, Kyoto University, Japan

**Correspondence:** Xuanming Su (suxuanming@jamstec.go.jp)

**Abstract.** Most recent cost-benefit Integrated Assessment Models have used only one reference scenario and focused on reducing mainly $CO_2$ emissions. This goal may not adequately account for the uncertainties arising from diverse socioeconomic developments and the potential for mitigating the effects of emissions of individual greenhouse gases, aerosols, and pollutants. We developed an Integrated Assessment Model framework by combining a socioeconomic module with a reduced-complexity

climate module. We represented the Shared Socioeconomic Pathways (SSP) scenarios by 1) calculating a new set of marginal abatement cost curves based on the most recent integrated assessment model, 2) creating a new SSP-dependent damage function based on process-based impact simulation results, and 3) extending the evaluation time to the year 2450. The cost-benefit analysis revealed that the SSP scenarios achieved various rates of control for emissions of individual greenhouse gases, aerosols, and pollutants. The result was diverse patterns of optimal temperatures, including maximum temperature achieved and stabi-

lized temperature by the end of the evaluation period. The model simulations showed the importance of distinguishing options for reducing emissions of greenhouse gases based on distinct socioeconomic growth scenarios. We also show an example of a long-term socioeconomic projection spanning several centuries as well as a variety of socioeconomic assumptions for assessing climate change policies with long-term consequences.

## 1   Introduction

Climate change researchers and policymakers use Integrated Assessment Models (IAMs) to investigate the intricate connections between socioeconomic systems and Earth's climate system. There are two categories of IAMs: cost-benefit IAMs (CB-IAMs) and process-based IAMs. CB-IAMs integrate a socioeconomic module with a simplified climatic module, whereas process-based IAMs model transformation processes that involve various activities that result in emissions as byproducts. CB-IAMs, such as the Dynamic Integrated Model of Climate and the Economy (DICE) (Nordhaus, 2008, 2013; Nordhaus and Sztorc, 2013; Nordhaus, 2014; Barrage and Nordhaus, 2024) or the comprehensive evaluation model based on it (Yang

and Sztorc, 2013; Nordhaus, 2014; Barrage and Nordhaus, 2024) or the comprehensive evaluation model based on it (Yang



**Table 1.** Main socioeconomic and climate-related indicators in 2100 for SSP reference scenarios and the DICE model.

| Scenarios | Population (Millions) | GDP (Trillions USD) | Industrial $CO_2$ (GtC/year) | $CO_2$ concentration (ppm) | Radiative forcing (Wm$^{-2}$) | Global temperature change (°C) |
|---|---|---|---|---|---|---|
| DICE2013R | 10167.4 | 511.41 | 28 | 858.3 | 6.9 | 3.9 |
| DICE2016R | 11069.3 | 908.51 | 19.3 | 826.4 | 6.8 | 4.1 |
| DICE2023 | 10534.3 | 839.22 | 24.3 | 842.9 | 7.4 | 3.8 |
| SSP1 | 6886.8 | 855.4 | 5.3 | 577.7 | 5.3 | 3.7 |
| SSP2 | 8960 | 726.5 | 16.0 | 752.2 | 6.9 | 4.6 |
| SSP3 | 12639.4 | 417.6 | 25.3 | 866.6 | 7.6 | 5.0 |
| SSP4 | 9263.3 | 493.2 | 7.8 | 617.1 | 5.5 | 3.7 |
| SSP5 | 7393.3 | 1448.6 | 21.9 | 849.8 | 7.7 | 5.1 |

et al., 2018; Glanemann et al., 2020; Brown and Saunders, 2020; Rickels and Schwinger, 2021; Koch and Leimbach, 2023) are typically used to evaluate economically optimal mitigation solutions. A CB-IAM assumes an ideal environment in which economies are willing to invest in reductions of current greenhouse gases (GHGs) in order to mitigate future climate damage. The optimal cost-benefit emission pathway can thus be established by weighing total temporal abatement costs against climatic

impacts.

Current CB-IAMs typically focus on a single reference emission scenario, which might not adequately capture the significant uncertainties surrounding socioeconomic development. For example, various versions of DICE models, including DICE2013R (Nordhaus and Sztorc, 2013; Nordhaus, 2014), DICE2016R (Nordhaus, 2017), or DICE-2023 (Barrage and Nordhaus, 2024) specify a reference scenario of relatively high radiative forcing (Table 1), close to the forcing projections of Representative

Concentration Pathway (RCP) 8.5 through 2100. The temperature increases for these reference scenarios range from 3.8 °C to 4.1 °C, depending on socioeconomic and climate-related assumptions. Among the fundamental determinants, population is assumed to grow relatively fast and exceed 10 billion by 2100, whereas outputs of total gross domestic product (GDP) are variable. The demographic and economic assumptions are important determinants of social welfare, which is maximized in the optimization processes of CB-IAMs. However, the overly uniform assumptions are not able to reflect the wide range of

possible future socioeconomic development scenarios. The Shared Socioeconomic Pathways (SSPs) scenarios (Moss et al., 2010; van Vuuren et al., 2012), which have been widely used and are discussed in the IPCC Sixth Assessment Report (IPCC, 2022), have defined five SSPs that reflect a broad spectrum of future challenges to mitigation and adaptation by considering potential future population sizes, economic activity, energy use, land use, technological progress, etc., as shown in Table 1. Here, we used SSPs to account for socioeconomic uncertainties in order to calibrate the socioeconomic module. We expected

the optimal emission pathways to be produced under different socioeconomic contexts.





The CB-IAMs address primarily reductions of $CO_2$ emissions or integrated GHG emissions. However, the SSP variations are crucial for determining the optimal emissions pathway because they affect the potential, ease, and costs of reducing emissions of GHGs as well as aerosols and pollutants (Su et al., 2017, 2018). It is therefore necessary to separate the possible reductions and costs of individual emissions for each SSP scenario. The economic concept of marginal abatement cost (MAC) is commonly used to calculate the cost of reducing one additional unit of emissions. Here, we estimate the SSP-dependent MAC curve, which reflects the link between the control of the rate of emission for each SSP and the related reduction cost. The MAC curve is used in the CB-IAMs to differentiate between the reduction potentials and costs of emissions of various GHGs, aerosols, and pollutants.

To illustrate the effects of climate change on human society, the DICE model uses a highly simplified damage function built from current surveys and estimates that include numerous susceptible sectors such as agriculture, forestry, fisheries, coastal real estate, and transportation (Nordhaus and Sztorc, 2013; Nordhaus, 2013). There would be no problem if different SSP projections were treated as future possibilities with relatively broad uncertainty using a single damage function. However, there are obvious differences of impacts among SSPs in specific areas such as occupational health costs and hydropower generation (Takakura et al., 2021). There is consequently a need to use an SSP-dependent damage function to distinguish between climate change damages under different SSP scenarios.

In the IPCC report series published since the 1990s, many emission pathways and socioeconomic scenarios (SRES, RCPs, SSPs, and so on) have been established to examine the effects of global temperature increases on Earth's ecosystems and human society. The majority of the time horizons for these future projections produced by IAMs in earlier studies was 2100. This horizon facilitates examination of likely future outlooks because these scenarios cover almost a century. The goal of the upcoming wave of research on global climate change and environmental issues is to foresee events beyond the year 2100 (Lyon et al., 2022; Meinshausen et al., 2023). To comprehensively represent the reaction of the climate to anthropogenic activities over medium and very long periods of time, the CB-IAM employs a projection horizon of 2450, which is comparable to that of the DICE model (Nordhaus, 2013). To make those projections, we aimed to expand all five SSPs to the year 2450 for the purpose of CB-IAM optimizations.

In this study, we developed a CB-IAM by combining a socioeconomic module with a Reduced-Complexity Module (RCM) (Su et al., 2017, 2018). We first improved the socioeconomic module by 1) estimating a new set of MAC curves based on the output of the most recent AIM/Hub V2.2 (Fujimori et al., 2017a, b, 2023) to simulate the mitigation potentials and costs of various emissions; 2) developing a new set of SSP-dependent damage functions using process-based impact simulation results (Takakura et al., 2021) to distinguish the climate impacts under different SSPs; and 3) extending the evaluation period of the SSPs to the year 2450 to capture the long-term climate response. We then revised the RCM module to make it suitable for use in the optimization process. The socioeconomic module and the RCM module were coupled to create the CB-IAM for determining the optimal climate change policies for SSP1-5. The flowchart for building the CB-IAM is presented in Fig. 1.



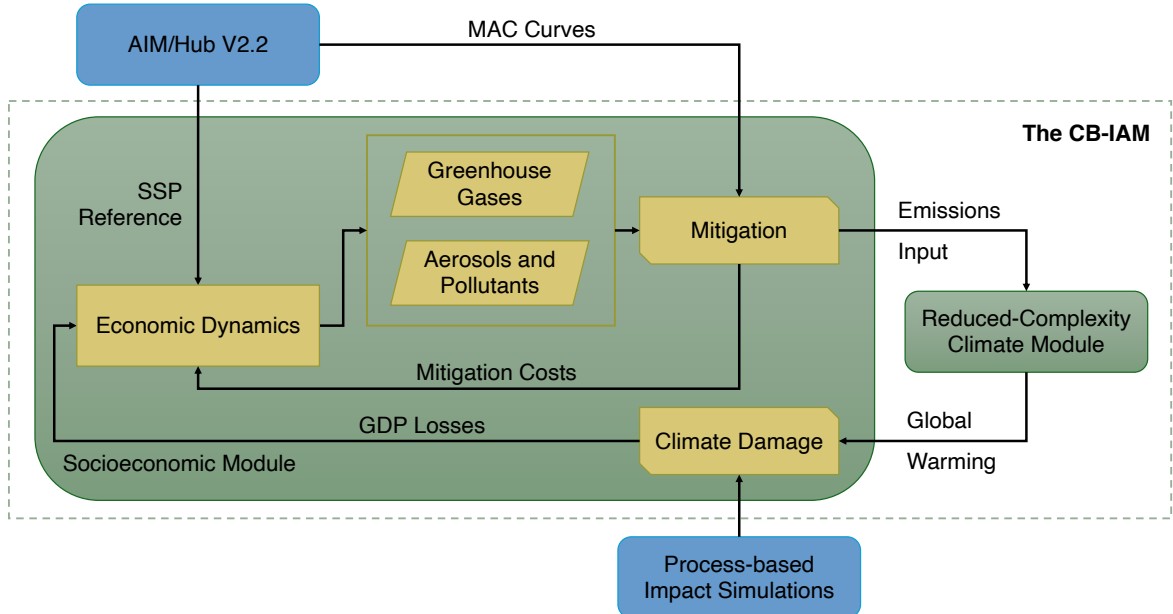

**Figure 1. Diagram of the modules for development of the cost-benefit Integrated Assessment Model (CB-IAM).** Colored areas show modules or their connected components. Black arrows represent the interactions between modules and components. The dashed line encloses the components of the CB-IAM. AIM/Hub V2.2 is taken from Fujimori et al. (2017a, b, 2023). The process-based impact simulations are from Takakura et al. (2021).

## 2   Methodologies

### 2.1   Estimation of MAC curves

We used MAC curves to compute the abatement costs given carbon prices and reduced emissions. To estimate MAC curves, we created a predetermined series of carbon price trajectories, from 0 USD (2010)/tCO$_2$ in 2020 to 100-500 USD (2010)/tCO$_2$ in 2100. The values within the time interval 2020-2100 were interpolated in a linear manner using the methodology used in previous studies (Su et al., 2017, 2018). In addition, we included a variety of low-carbon-price scenarios, ranging from 3 to 70 USD (2010)/tCO$_2$ in 2100 (Fig. A1), to highlight the potential long-term emission reductions from low-carbon prices. We

produced trajectories of emissions for individual GHG emissions, aerosols, and pollutants using the AIM/Hub V2.2 (Fujimori et al., 2017a, b, 2023) with these carbon price paths as constraints.

A two-term function (Su et al., 2017, 2018) was used to characterize the relationship between rates of control of emissions, defined as the annual abated emissions divided by total emissions, and carbon prices based on the generated emissions trajectories.



$$p_c(t, gas) = \theta_{1,gas}\mu(t, gas)^{\theta_{2,gas}} + \theta'_{1,gas}\mu(t, gas)^{\theta'_{2,gas}} \qquad (1)$$

where $p_c(t, gas)$ represents the carbon price in year $t$ for emission $gas$, $\mu(t, gas)$ denotes the rate of control of specified emissions, and $\theta_{1,gas}$, $\theta_{2,gas}$, $\theta'_{1,gas}$ and $\theta'_{2,gas}$ are estimated parameters (Table A1-A5). Equation 1 describes the economic behavior in such a way that the carbon price increases gradually when the rate of control is relatively low and rapidly when more control is required because of the difficulty of making additional reductions (Su et al., 2017, 2018).

Here, the control rate $\mu(t, gas)$ was applied to the total sum of several sectors based on their respective emissions (Table A6). We performed a sensitivity test on each emission over multiple sectors. The test results showed that changing the carbon price had little effect on reductions of specific emissions in certain sectors. For example, land-use emissions were determined primarily by how different kinds of land covers were used in the future, and these decisions were rather insensitive to carbon prices. Furthermore, clean-air policies that limited emissions of aerosols and pollutants were explicitly considered in AIM/Hub

V2.2, and they were also insensitive to carbon prices. Reductions of land-use-related emissions were therefore not considered in calculating the $\mu(t, gas)$ of GHG emissions. For aerosols and pollutants, reductions of emissions due to land use, the energy sector (commercial, residential, and industrial usage), industrial processes, and waste were not considered in the calculations (Table A6).

Fig. 2 shows the estimated MAC curves for various GHG emissions, aerosols, and pollutants. The relationship between

rates of emission control and carbon prices was determined by the SSP scenarios and associated emissions. There were several noteworthy aspects of the results. First, at the same control rates, our estimated carbon costs were often lower than the MAC curve used in DICE2023 (Barrage and Nordhaus, 2024) because of an improvement in the AIM/Hub V2.2 that reflected the lower costs of renewable energy sources such as solar and wind that have been installed in recent years. Second, among the SSP scenarios, SSP4 and SSP5 had high control rates with much lower costs than the other scenarios, and SSP3 had the highest cost.

Achieving a high control rate or a negative emission was difficult in the SSP3 scenario. Third, the reduction potentials for non-$CO_2$ GHGs such as $CH_4$ and $N_2O$, as well as aerosols and pollutants, were strongly linked to the socioeconomic assumptions underlying each SSP scenario. Modeling the reductions independently for each emission would show the disparities between mitigation potentials and costs.

### 2.2 The socioeconomic module

We used a neoclassical economic module akin to the one used in the DICE model to simulate future economic development in terms of variables such as population, production output, investment, and consumption. In this approach, economies invest in current mitigation technologies with the aim of diminishing forthcoming negative natural capital that may arise because of GHG emissions. In order to accommodate the incorporation of the SSPs and the RCM, we made certain modifications to the economic module. For instance, we changed the time step from five years to one year, extended the SSPs to the year 2450,

implemented a new SSP-dependent damage function, and accounted for reductions not only in the individual GHG emissions, but also in the aerosols and pollutants, as illustrated in the following subsections.







**Figure 2. Marginal abatement cost (MAC) curves for the SSP1-5 scenarios.** Colored points show sensitivity data relating the rates of control of emissions to carbon prices based on AIM/Hub V2.2. Colored lines reflect the MAC curves estimated with Eq. 1. Note that for GHG emissions, reductions of land-use-related emissions are not covered; for aerosols and pollutants, reductions of emissions from land use, the energy sector (commercial, residential, and industrial use), industrial processes, and waste are not included (Table A6). The gray line and range on the $CO_2$ panel indicate the DICE2023 MAC curve.



### 2.2.1 Reference scenario population and GDP

We extrapolated the population forecasts to the year 2450 based on our previous studies (Su et al., 2017, 2018). The populations were assumed to adhere to the trends outlined in the SSPs (from the output of AIM/Hub V2.2) for the years after 2100 and to stabilize after 2150. As depicted in Fig. 3**a**, our population assumptions encompassed a broad spectrum of future population levels, ranging from relatively low values of 6 billion to high values of 14 billion. These values correspond to the levels projected by the SSPs for the year 2100 and may be compared to the population assumed in DICE2023, which is approximately 10,825 million in the long run.

For the GDP, we extended the values beyond 2100 based on the growth rate of GDP per capita ($\omega$(t)) in 2100 as follows:

$$\omega(t) = \omega(t_{2100}) \cdot \exp(-\frac{t - t_{2100}}{\tau_\omega}) \tag{2}$$

where $\omega(t)$ and $\omega(t_{2100})$ denote the GDP per capita growth rate in year $t$ and 2100, respectively, and $\omega(t_{2100})$ is computed using the output of the AIM/Hub V2.2. The decay time, $\tau_\omega$, was defined to be 50 years. To prevent improbably large growth of the estimated GDP, the per capita growth rates of GDP were assumed to decline after 2100. The GDP per capita projections for the SSPs are shown in Fig. A2. In the neoclassical economic growth module, labor (population) and capital, along with the total level of productivity, are used as inputs to a Cobb-Douglas production function to determine the final gross output. We initially estimated the target production output by multiplying the population and GDP per capita calculated in accord with the trends derived above. Next, we used an inverse version of the CB-IAM that accepted the target production output as a constraint in the reference scenario to estimate the total level of productivity (Fig. A3). The GDP for the SSP reference scenarios were obtained using this total level of productivity (Fig. 3**b**). The total GDP was substantially less in the long run than in the DICE 2023 model. However, by the end of the evaluation period, the GDP per capita in the SSP5 scenario was nearly in line with the DICE2023 model, but the GDP per capita was lower than in the DICE2023 model in the other SSPs to various degrees (Fig. A2).

### 2.2.2 Anthropogenic emissions for reference scenarios

The economic module was expanded to completely capture the abatement potentials for various anthropogenic emissions, i.e., $CO_2$, $CH_4$, $N_2O$, CO, VOC, $SO_x$, $NO_x$, BC and OC (the mitigation of halogenated gases was not included because of data availability). Based on the growth trajectories of the SSP scenarios before 2100, the emissions of the reference scenario were extended beyond that year. We first used the output of AIM/Hub V2.2 to compute the emission intensity, defined as the amount of emissions produced per unit of GDP, by the year 2100. As was done in the DICE2023 model, we projected an exponential reduction in carbon intensity of -0.1% per year in the rate of increase of emission intensity for the forecast beyond 2100. We also adjusted the emission intensity of SSP2 and SSP3. For SSP2, we assumed a rate of exponential decline of 1.5 times higher than other scenarios to ensure that SSP2 would reflect a medium emission scenario over time. As with SSP3, we assumed a rate of annual change of -1% without taking into account the exponential form to prevent excessively high emissions in the future that could have resulted in a $CO_2$ concentration above 3000 ppm, which has rarely been seen in current model simulations.





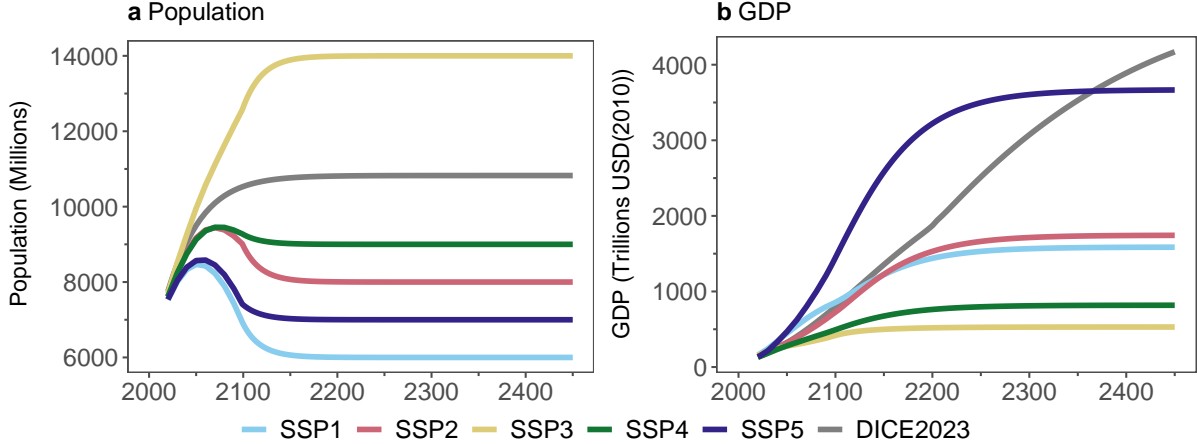

**Figure 3. Assumptions on (a) population and (b) GDP.** The population is assumed to stabilize after 2150 (Su et al., 2017, 2018). Colored lines show the SSP projections, and the gray line reflects the assumption of DICE2023.

Fig. 4 shows the intensity of the emissions. As stated in the SSP narratives, with the exception of SSP3, the majority of the emission intensities were predicted to decrease rapidly by 2100 and stabilize at a very low level by the end of the evaluation period. In the case of SSP3, however, the $CO_2$ emission intensity was essentially unchanged by 2100. Furthermore, based on the assumptions, the intensity of emissions other than $CO_2$ emissions were much larger for SSP3 than for the other SSPs, both before and after 2100. As a result, the future emissions indicated that while $CO_2$ emissions were comparatively low for SSP1 and SSP4, the $CO_2$ emissions were especially large for SSP3 and SSP5 (Fig. 5). The emissions of other GHGs such as $CH_4$ and $N_2O$ were based on trends predicted by the SSPs prior to 2100. With the exception of SSP3, which continued to show relatively high rates of emission, emissions of most of the aerosols and pollutants for the other SSPs fell quickly over the first half of the evaluation period.

### 2.2.3 Mitigation of GHG emissions and aerosols and pollutants

In this study, we separately simulated the mitigation of GHGs, aerosols and pollutants (Table A6). For the emissions that could be abated, the level of the emission after abatement, $E_{abate}(t, gas)$, was given by

$$E_{abate}(t, gas) = \sigma(t, gas) Y_{gross}(t) (1 - \mu(t, gas)) \tag{3}$$

where $\sigma(t, gas)$ is the emission intensity per thousand USD (2010), $Y_{gross}(t)$ denotes the gross output in trillions of USD (2010), and $\mu(t, gas)$ represents the rate of control of the emissions (vide supra).

The non-abatable emissions, $E_{non-abate}(t, gas)$, was equated to the associated reference level $E^{ref}_{non-abate}(t, gas)$:

$$E_{non-abate}(t, gas) = E^{ref}_{non-abate}(t, gas) \tag{4}$$





**Figure 4. Emission intensity for the reference scenarios of the SSPs.** Colored lines show the SSP projections. The values before 2100 are estimated based on output of the AIM/Hub V2.2.




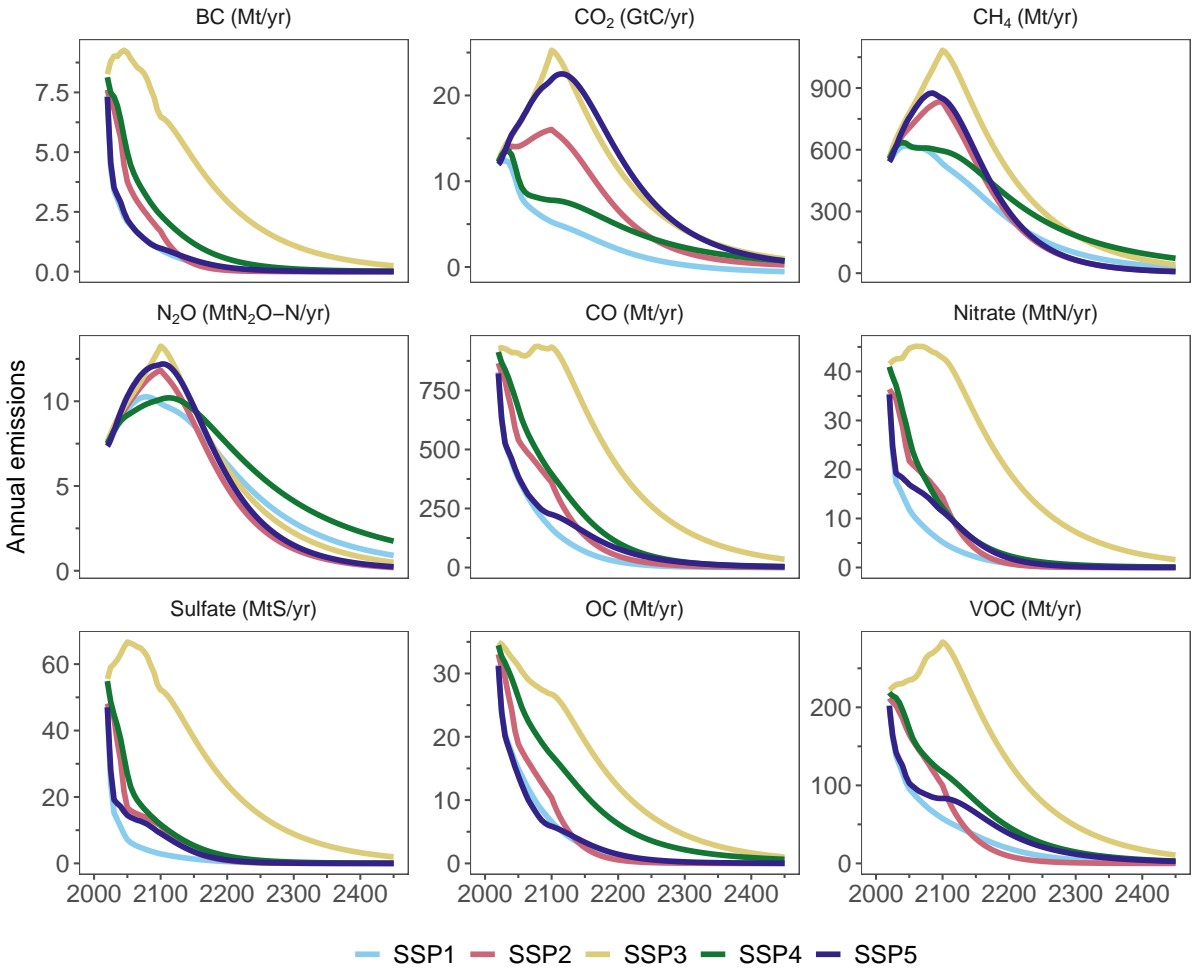

**Figure 5. Emissions of GHGs, aerosols, and pollutants for the reference scenarios of the SSPs.** Colored lines show the SSP projections.

The total emission $E(t, gas)$ for the SSPs was calculated by adding the abatable emissions $E_{abate}(t, gas)$ and the non-abatable emissions $E_{non-abate}(t, gas)$:

$$E(t, gas) = E_{abate}(t, gas) + E_{non-abate}(t, gas) \qquad (5)$$

By integrating the MAC function (Fig. 2), the mitigation cost $\Delta_{abate}(t, gas)$ in trillions of USD (2010) was obtained as

$$\Delta_{abate}(t) = \frac{1}{1000} Y_{gross}(t)\nu(t)\sigma(t, gas) \left( \frac{\theta_{1,gas}\mu(t,gas)^{\theta_{2,gas}+1}}{\theta_{2,gas}+1} + \frac{\theta'_{1,gas}\mu(t,gas)^{\theta'_{2,gas}+1}}{\theta'_{2,gas}+1} \right) \qquad (6)$$



where $\nu(t)$ is the carbon price adjustment factor due to technological advancements. We estimated the abatement costs for the emissions of $CO_2$, $CH_4$ and $N_2O$ and calculated the overall abatement cost by adding them all together.

### 2.2.4 Climate change damages

As with the DICE model, we used a damage function to describe the economic damages or consequences of climate change:

$$\Delta_{gross}(t) = Y_{gross}(t)\left(a_1 T(t) + a_2 T(t)^{a_3}\right) \tag{7}$$

The variables $\Delta_{gross}(t)$ and $Y_{gross}(t)$ represent gross climate damage and gross production output in trillions of USD (2010), respectively. The tuned parameters were $a_1$, $a_2$ and $a_3$. In this work, we re-estimated the SSP-dependent damage functions based on process-based impact simulations (Takakura et al., 2021). Fig. A4 shows that the damage function used in this study generally had a larger impact on GDP losses than previous DICE versions, yet DICE2023 fell within our range of

estimates.

### 2.3 The simple climate module

The Simple Climate Model for Optimization version 3.3 (SCM4OPT v3.3) (Nicholls et al., 2020; Su et al., 2022) is a radiative forcing and global temperature simulation model that uses a full suite of GHG, aerosol, and pollutant emissions as well as land-use albedo as input. The SCM4OPT v3.3 includes a carbon cycle module for estimating atmospheric $CO_2$ concentrations;

an atmospheric chemistry module for simulating the atmospheric evolution of the emissions of non-$CO_2$ GHGs, aerosols and pollutants; and a climate module for estimating the temperature response to the associated radiative forcing. For use in long-term optimization, the SCM4OPT v3.3 was revised to 1) depict the carbon cycle using an impulse response function (Barrage and Nordhaus, 2024) and 2) switch to a one-year time step from a bimonthly time step. We used an equilibrium climate sensitivity range of 2 °C to 5 °C (Forster et al., 2021) to take into account climate-related uncertainty.

The simple climate module simulated the increase of global mean temperature above the preindustrial level, which has been caused primarily by increased atmospheric $CO_2$ concentrations due to anthropogenic and natural $CO_2$ emissions. We placed a lower bound on the preindustrial $CO_2$ concentration, which was 284 ppm in 1850, to avoid possible excessive cuts in the distant future. Based on this premise, no additional $CO_2$ reductions would be carried out if future concentrations reached 284 ppm. The implication was that a negative global mean temperature increase above preindustrial levels was implausible in our

cost-benefit analysis.

### 2.4 The reproducibility of the AIM/Hub V2.2 output

The CB-IAM simulated conditions from 1850 to 2450. We acquired the time series for the historical period (1850-2019) from existing datasets. For example, the population data were obtained from a UN report (United Nations Secretariat, 1999) and UN Population Databases; the gross output was estimated from Maddison (2007), and the GHGs, aerosols and pollutants were

obtained from the Community Emissions Data System (Hoesly et al., 2018), EDGAR v7.0_GHG 1970-2021, and EDGAR





v6.1_AP 1970-2018 (Crippa et al., 2021, 2022). The variables within the SSP projected period, 2020-2100, were derived from the output of AIM/Hub V2.2, whereas those beyond 2100 were assumed and extended in this study. Table A7 shows the primarily variables.

We used the MAC curves for each individual emission to replicate the mitigation potentials for the future period. We conducted a calibration test and used the carbon price pathways as constraints to generate the paths of control rates. To assess the reproducibility, we compared the control rates for the individual emissions between our IAM and the AIM/Hub V2.2. The primary trends for the emission mitigations were modeled in our IAM for each SSP (Figs. A5-A9). The simulations of $CO_2$ mitigation in SSP2, SSP3, and SSP5 outperformed those in other SSP scenarios. For all SSPs, the simulation was generally more accurate for reductions of GHGs than for reductions of aerosols and pollutants. The reason was that the estimated non-abatable emissions contributed a greater proportion of the emissions of aerosols and pollutants. Our model did not take into account the clean air rules that regulate these emissions. In addition, the near-term projections of most emissions of SSPs were characterized by greater uncertainty, mostly because of the use of time-independent MACs. The use of time-independent MACs resulted in a consistent control rate for a given carbon price. Nevertheless, the control rate derived from the output of the AIM/Hub V2.2 was variable when carbon prices were low (as shown in Fig. 2). The result was a greater uncertainty in the simulation for the near future. Because of these uncertainties, it was important to exercise caution when interpreting the optimization outcomes generated by this CB-IAM.

## 3 Results

We ran simulations for two scenarios for each SSP: the reference scenario, which called for no future mitigations, and the optimal scenario, which balanced the present values of the costs associated with mitigation with the present values of climate damages under the SSP projections and related extensions. To further illustrate how the optimal emission pathways would develop under the SSP assumptions, we also conducted three sensitivity tests: 1) a comparison was made between mitigation of only $CO_2$ as opposed to the scenario with full mitigation of all available emissions; 2) a comparison was made between the use of the new damage function and that used in the DICE2023 model; and 3) a cross-sensitivity test was conducted regarding population, the level of total productivity, and MACs to highlight the most important factors that influenced the long-term stabilized temperature and optimal climate policies.

### 3.1 Global temperature change

Fig. 6 shows the global temperature increases associated with the SSPs for the reference and optimal scenarios. For the reference scenarios, there were significant temperature increases for SSP3 and SSP5, and the temperature increase was smallest for SSP1 in the long run. The temperature increases in the reference scenarios reflected climate change as a result of assumptions about SSP emissions and long-term extensions. The SSP3 scenario led to both the highest optimal temperature, namely the maximum temperature under optimal conditions, and the highest stabilized temperature, most likely because of the large emissions and the difficulty in reducing those emissions, as indicated by the calculated MACs. The optimal scenarios for SSP1,



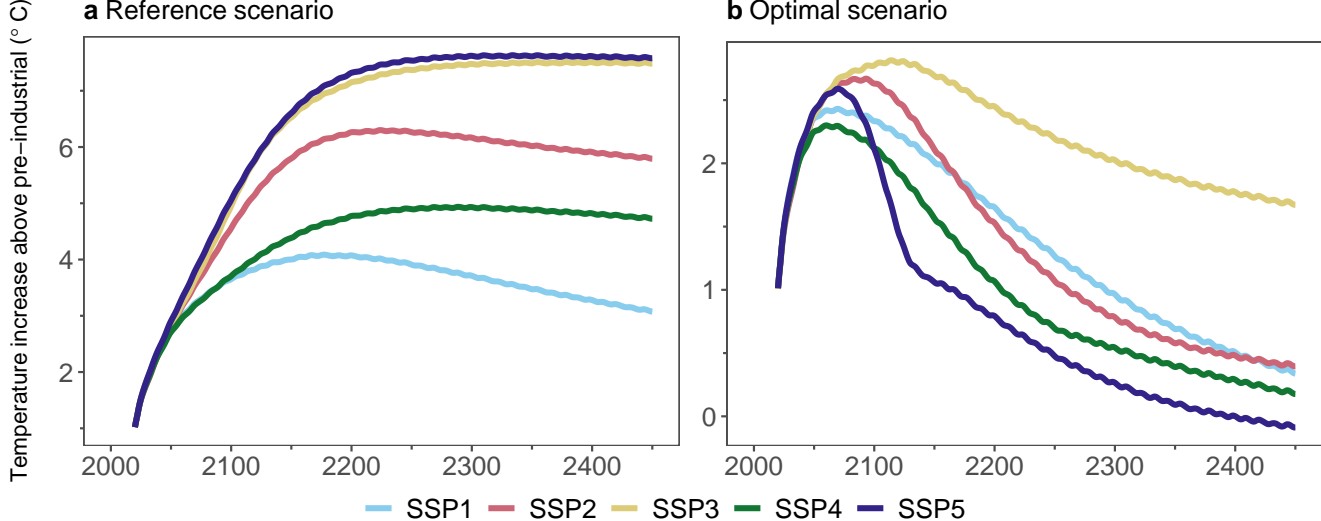

**Figure 6. Global temperature changes for the SSPs in the (a) reference scenario and the (b) optimal scenario.** Colored lines show the SSP projections.

SSP2, and SSP4 all had similar stabilized temperatures at the end of the assessment period. However, SSP2 had a higher optimal temperature than the other two because it assumed comparatively high emissions in the near term. It is important to note that, whereas SSP5 had a high temperature in the reference scenario, it had the lowest stabilized temperature in the optimal scenario, possibly because of the cheaper cost of reducing the same percentage of emissions, as presented in the MACs, among other reasons.

## 3.2 Mitigation of emissions of GHGs, aerosols and pollutants

The optimal scenario, although not feasible in reality, serves as a benchmark for evaluating the effectiveness of various climate change measures (Nordhaus and Sztorc, 2013). Fig. 7 shows the emission control rates for the SSPs. In the case of $CO_2$, all SSPs with the exception of SSP3 resulted in notably high maximum control rates that ranged from 1.36 in SSP1 to 1.53 in SSP5. The low maximum control rate in the case of SSP3 was attributable to the MAC, wherein the carbon price increased significantly as the control rate approached 1. SSP5 stood out by offering an early and substantial reduction in carbon emissions compared to other SSPs because of its comparatively low carbon price required to achieve this reduction. Nevertheless, the control rate of SSP5 underwent a continuous decline during the latter part of the evaluation period, primarily because of the lower limit of the pre-industrial $CO_2$ concentration. The control rates for emissions of other GHGs, aerosols and pollutants were all below 1, and the majority of them remained relatively constant after 2100 when subjected to high carbon costs.

The total emissions included abatable emissions, some of which were lowered in the scenarios, as well as emissions that could not be decreased in the models, such as land-use-related emissions and other emissions linked to carbon-price-insensitive sectors (Fig. 8). The SSP5 scenario led to strongly negative $CO_2$ emissions in the 2110s because of its early, high rate of $CO_2$




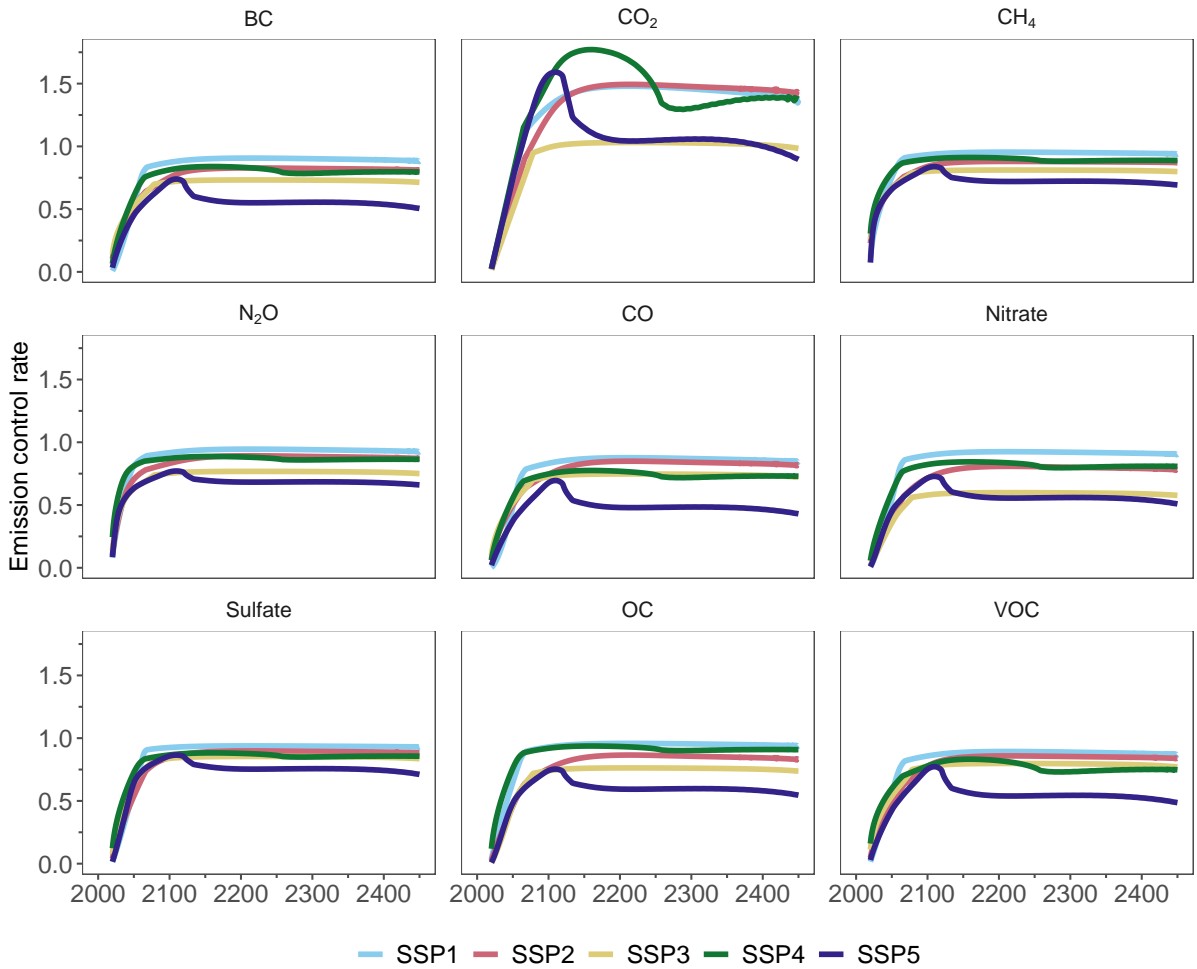

**Figure 7. Control rates of optimal scenarios.** Colored lines show the SSP projections.

control. SSP5 also exhibited relatively low emissions of $CH_4$, $N_2O$, aerosols and pollutants. The $CO_2$ emissions in all the SSPs stabilized at around zero in the second half of the assessment period. However, the $CO_2$ emissions decreased much more slowly in the case of the SSP3 scenario than the other scenarios in the first half of the evaluation period. In addition, most of the non-$CO_2$ emissions declined at a slower rate in SSP3 than in the other SSPs. The comparatively low control rate and high

net SSP3 emissions resulted in a high optimal temperature as well as a high stabilized temperature. Because of the contribution of non-abatable emissions to the $CH_4$ and $N_2O$ emissions, emissions of $CH_4$ and $N_2O$ increased before 2100, although some of the abatable emissions had been eliminated by then. After 2100, however, both the abatable and non-abatable emissions were assumed to decline in the reference scenarios because of technological developments. The resulting optimal emission pathways of $CH_4$ and $N_2O$ therefore declined even further for all the SSPs.





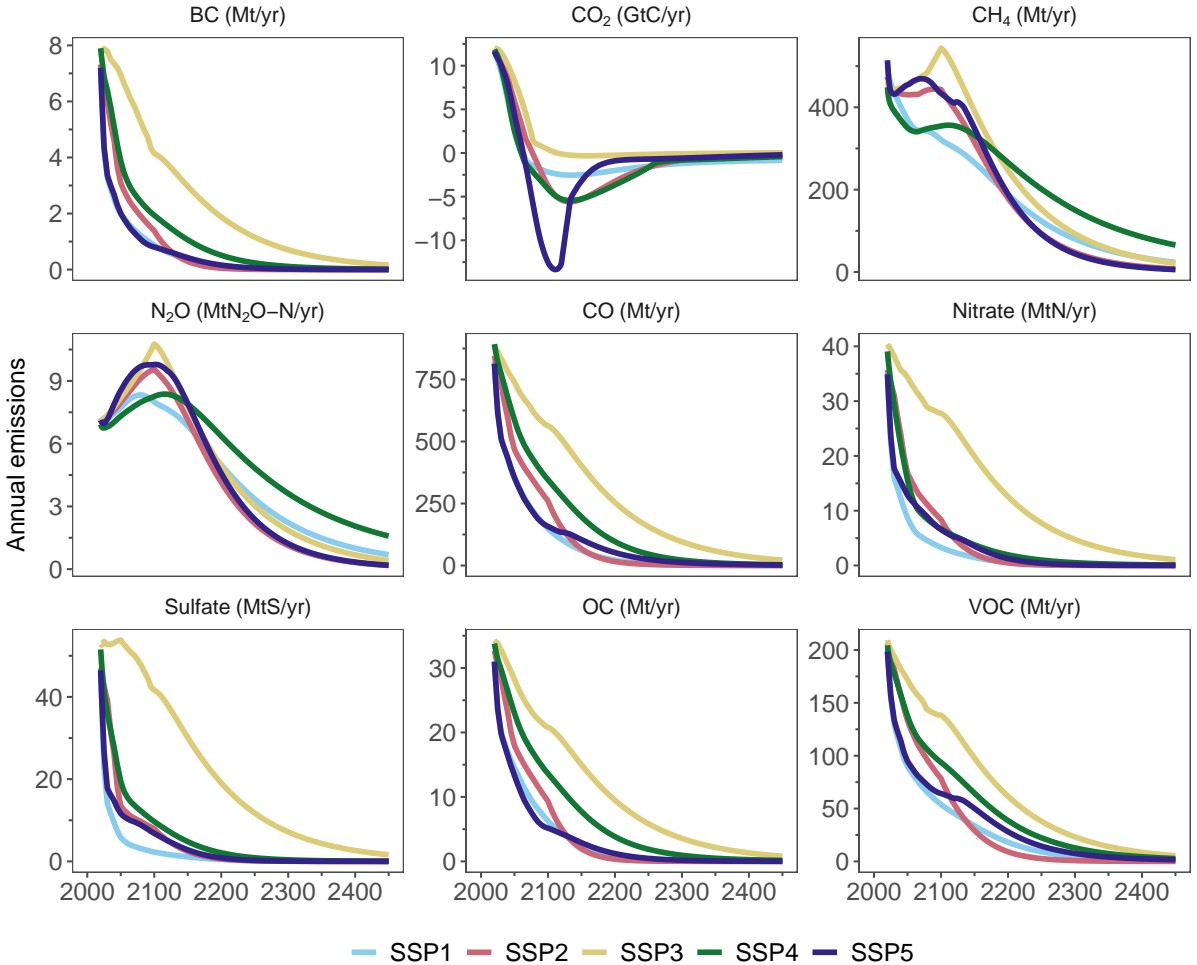

**Figure 8. Optimal emission pathways.** Colored lines show the SSP projections.

## 3.3 Mitigation costs and climate damages

The optimal climate policies are those that strike a balance between the costs of mitigation and the effects of climate change. The rates of control of $CO_2$ emissions, which will reach their maxima in the first half of the assessment period (Fig. 7), were the primary determinants of carbon prices. Consequently, as Fig. 9**a** illustrates, the highest carbon prices occurred between 2100 and 2200. The carbon prices were relatively low in SSP3 and SSP4 and relatively high in SSP1 and SSP2. Because it was easier to reduce $CO_2$ emissions and other non-$CO_2$ emissions in SSP5, the price of carbon in SSP5 reached a maximum in 2115, slightly earlier than in the other scenarios. After 2115, the price continued to decline, in part because of the constraint that the $CO_2$ concentration not decrease below the preindustrial level, among other reasons. Even if carbon prices were assumed to be modest, SSP5 led to the highest abatement costs because of the significant quantity of GHG emissions that were removed




under the optimal scenario (Fig. 9**b**). Depending on the associated carbon prices and the quantity of GHG emissions that had

to be decreased in order to achieve the goals of the optimal climate policies, SSP1 and SSP4 led to minimal abatement costs.

Climate damages were calculated as a fraction of gross economic production (trillions of USD) based on the increase in global mean surface temperature over preindustrial levels. On the one hand, higher temperatures may lead to a higher percentage of climate damage. On the other hand, absolute climate damages are also affected by gross economic production, and a larger gross economic production results in greater absolute climate damage with the same temperature increase. The

annual climate damage in the SSP5 scenario was thus greatest around the year 2100 and rapidly decreased afterward (Fig. 9**c**). SSP2 caused significant annual climate damage, most likely because of its relatively high optimization temperature and gross production output. Climate damages for SSP3 were moderate and diminished slowly during the latter half of the evaluation period because fewer emissions were reduced, and the ensuing temperature increases were generally higher in SSP3 than in the other scenarios. Finally, the net production output, which was affected by mitigation costs, climate damages, and total

economic activities, followed the same patterns as the gross production output (Fig. 9**d**). However, the GDP losses varied significantly depending on the socioeconomic assumptions. The GDP losses in SSP3 were as large as 5.4% but were at most 2.1% for the optimal scenarios of SSP1.

## 4   Discussion

### 4.1   Impacts of climate sensitivity

The fact that our model defaulted to a climate sensitivity of 3 °C was consistent with the IPCC Sixth Assessment Report (Forster et al., 2021). This metric addresses climate-related uncertainty. Fig. 10 displays the results for several climate sensitivity values ranging from 2 °C to 5 °C. Climate sensitivity has a considerable impact on the value of the optimal temperature. For example, a climate sensitivity of 5 °C resulted in a larger maximum temperature than a climate sensitivity of 2 °C. The climate sensitivity also influenced the value of the stabilized temperature at the end of the evaluation period was apparent in the comparative

behavior of SSP2, SSP3, and SSP4. The SSP1 temperature stabilized at 0.6 °C by the end of the evaluation period, possibly because of the complex mix of the remaining GHG emissions, aerosols and pollutants. SSP5 stabilized at a low level, despite the wide range of climate sensitivity. It is important to highlight that other climate-related uncertainties existed, such as model uncertainties from the carbon cycle, atmospheric physical and chemical processes, and external natural factors, that may have influenced the results of the optimization.

### 4.2   The discount rate

The discount rate is well known to have an important role in cost-benefit evaluations of long-term issues such as climate change (Weitzman, 1994, 2001; Dasgupta, 2008; Nordhaus, 2017). A higher discount rate reduces the present value of future social welfare and thereby reduces the influence of future generations on current policy decisions. For example, a higher discount rate reduces current carbon prices or the social cost of carbon (Yang et al., 2018; Emmerling et al., 2019). In this study, we

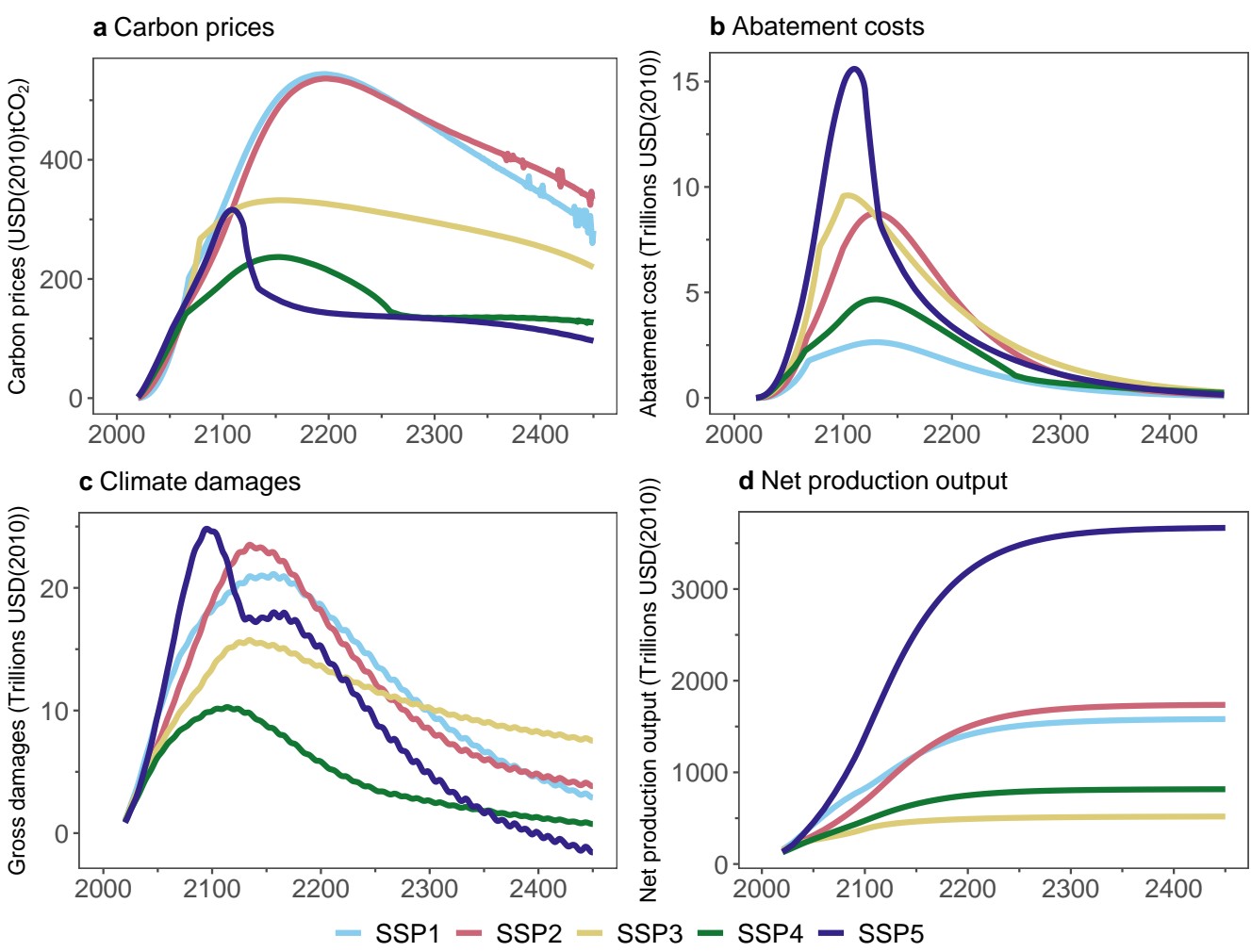

**Figure 9. (a) Carbon prices, (b) mitigation costs, (c) climate damages, and (d) net production outputs for the SSPs.** Colored lines show SSP projections.





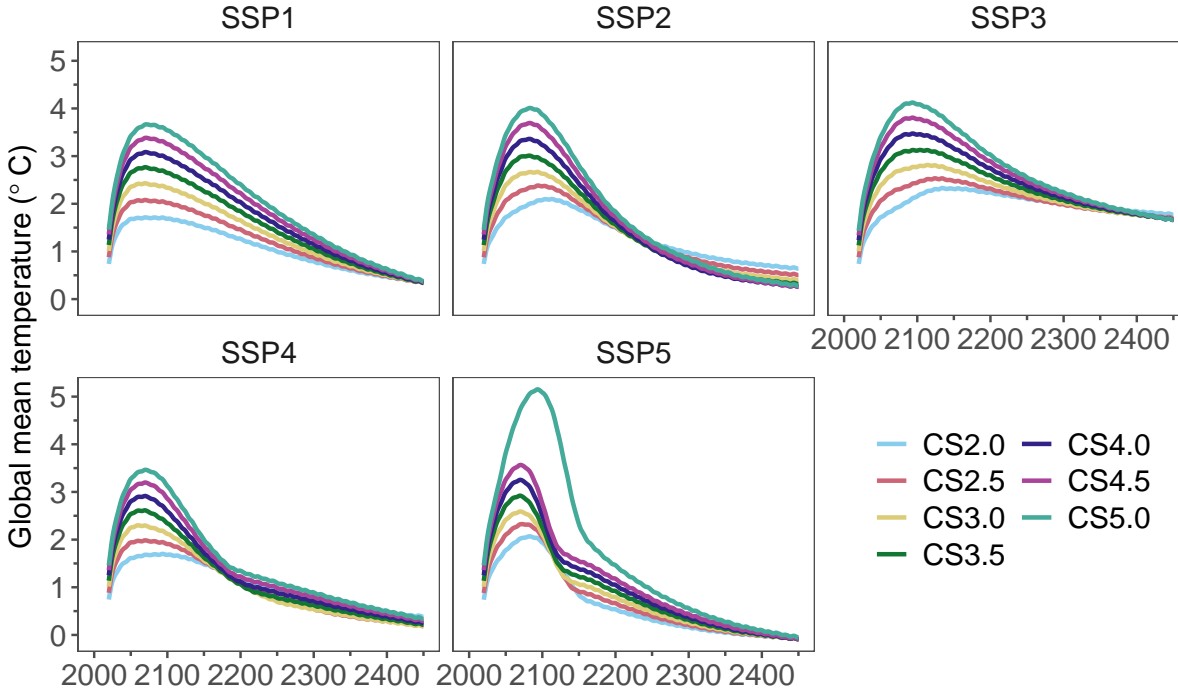

**Figure 10. Climate sensitivity test.** Colored lines show the SSP projections for climate sensitivity.

assumed the same parameters for all SSPs, namely, a pure rate of social time preference of 1.5% per year and an elasticity of marginal utility of 1.45. The result was an average (2020-2100) discount rate of 2.3% in SSP3 to 5.9% in SSP5 compared to 3.9% in DICE2023 during the same period (Fig. A10). The discount rate depended on the assumed growths of total factor productivity and population. As a result, the current climate policies of SSP5 were less influenced by future climate impacts caused by emissions, but the near-term climate policies of SSP3 were more influenced by future climate impacts.

## 4.3 Sensitivity test using solely $CO_2$ emissions and SSP-dependent damage functions

The CB-IAM distinguished between specific emission mitigations via different SSPs and employed an SSP-dependent damage function. We conducted a sensitivity test to determine how the optimal temperature would vary if mitigation were limited to $CO_2$ emissions ("SSP-dependent damages with solely $CO_2$") or if the damage function of DICE2023 was applied ("DICE2023 damages"), as illustrated in Fig. 11. The results of this sensitivity test were compared to the optimal scenarios ("SSP-dependent damages") in this study. The scenarios of SSP-dependent damages with solely $CO_2$ and SSP-dependent damages clearly had different optimal temperatures. The optimal temperature was higher for scenarios that included SSP-dependent damages due to solely $CO_2$ than scenarios of SSP-dependent damages. For example, the difference in the SSP3 scenario reached 0.52 °C. The implication was that mitigation measures for available GHGs, aerosols and pollutants might reduce optimal temperatures more than mitigation measured that involved solely $CO_2$. The explanation is that a major portion of the reduced $CH_4$ and



$N_2O$ emissions were byproducts of $CO_2$ mitigation, and the same carbon price could reduce more $CO_2$-equivalent emissions. For stabilized temperatures, however, both scenarios of SSP-dependent damages with solely $CO_2$ and SSP-dependent damages finally reached a similar level of temperature increase, mainly because the abatable emissions of GHGs, aerosols and pollutants were almost removed. In that case, most of the non-$CO_2$ emissions in the scenarios that included SSP-dependent damages with solely $CO_2$ would also be cut because of the assumptions made about technological change. The optimal temperatures differed

slightly between the DICE2023 and SSP-dependent damage scenarios. The implication was that the SSP-dependent damages estimated in this study may have had a relatively small impact on the optimal temperature, partly because there was little difference in climate damages when the temperature increase was less than 3 °C (Fig. A4), which also corresponded to the optimal temperature range, despite the fact that economic losses were significantly greater when global temperatures were warmer. However, there was a small difference in the stabilized temperatures between the cases with DICE2023 damages and

those with SSP-dependent damages. The sensitivity results indicated the importance of considering all the available emissions and using SSP-dependent damage functions for cost-benefit analysis.

## 4.4    Indicators affecting optimization scenarios

Our optimization results suggested that the SSP scenarios could generally stabilize at a low temperature by the end of the evaluation period. In particular, SSP5 might stabilize at a temperature difference as low as 0 °C. To determine the significance of

potential indicators such as population size, MACs, and the total productivity factors that contributed the most to gross production output, we performed a cross-sensitivity test in which we changed the potential indicators based on other SSP scenarios for a given scenario. Whereas such a sensitivity test may lack consistency among indicators, it can simulate a hypothetical experiment to determine how essential a possible indicator may be to achieving optimal outcomes. Fig. 12 depicts the stabilized temperature of the SSPs using several indicators. First, as illustrated in Fig. 12**a**, the stabilized temperature depended primarily

on socioeconomic assumptions. The stabilized temperatures tended to cluster around a specific value, and there was a range of values for each SSP. However, there was a low stabilized temperature in SSP3 under certain conditions, whereas the stabilized temperatures were relatively high in SSP5 and depended on the indicator assumptions. In general, smaller reduction costs could result in a lower stabilized temperature, as evidenced by the SSPs using the MACs from SSP4 and SSP5, which had relatively low carbon prices and high control rates. The impacts of the population and total productivity factors were ambiguous because

both indicators affected gross production output according to our assumptions about the Cobb-Douglas production function. The results of a cost-benefit analysis showed that lowering reduction costs could lead to a lower temperature in the long run, even in the case of a high emission scenario like SSP3.

Fig. 12**b-d** shows scatter plots of stabilized temperatures versus population, GDP, or GDP per capita. All SSPs using the same indicators are represented in the panel by the same symbol. In this way, the analysis focuses solely on the indicators,

regardless of the socioeconomic assumptions of the SSPs. The results indicated that the effects of population on stabilized temperatures were ambiguous because various amounts of stabilized temperature can occur at different population levels, as illustrated in Fig. 12**b**. Nonetheless, higher GDP or higher GDP per capita tended to reduce the stable temperature (Fig. 12**c-d**). The implication was that high GDP growth may help to keep the temperature at a low level in the long run. This pattern can be






**Figure 11. Temperature increases for the sensitivity scenarios.** Colored lines show the SSP projections for the sensitivity scenarios of SSP-dependent damages, SSP-dependent damages with solely $CO_2$, and DICE2023 damages.



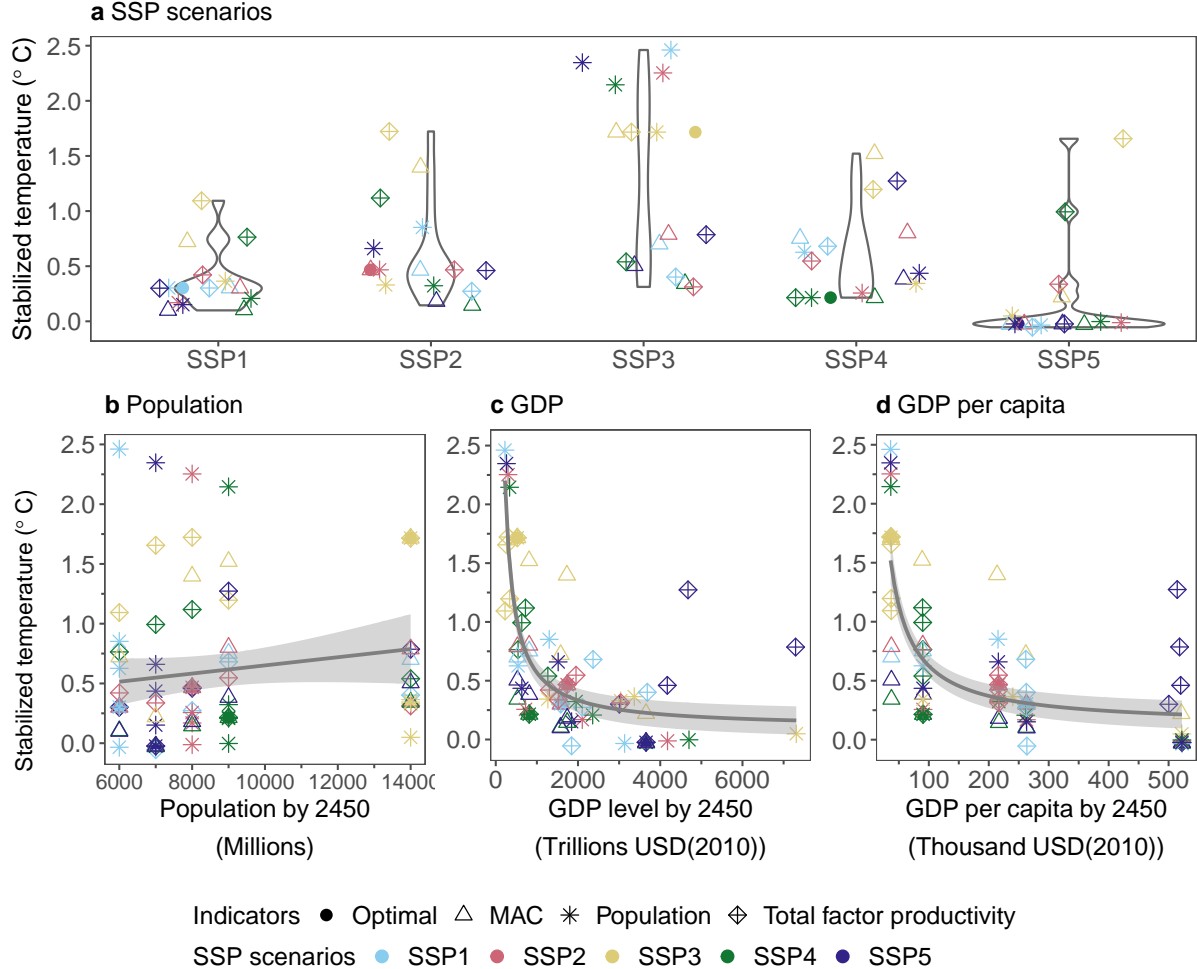

**Figure 12. A sensitivity test with respect to the stabilized temperature: (a) SSP scenario, (b) population, (c) GPD, and (d) GDP per capita.** The color of a point indicates the SSP scenario, and the shape of a point denotes the indicator used for the sensitivity test, either the population, total factor productivity, or MAC.

explained intuitively by assuming that wealthier economies are more willing to pay for climate mitigation in order to reduce

potential future climate damage. The assumptions for SSP5 also showed that, despite high emissions, reduced mitigation costs and increasing GDP growth can lead to a low, long-term, stable temperature.

## 5   Limitations and future studies

Our findings should be evaluated with caution for two reasons. First, the SSP projections to 2100 assumed a framework that included the main socioeconomic assumptions regarding population, GDP development, and industrial and land-use-related




emissions based on AIM/Hub V2.2, whereas regional socioeconomic development and detailed technologies beyond 2100 were ignored because the information necessary for such a long-term projection was lacking. We attempted to construct a consistent framework beyond 2100 using SSP definitions, which are only a small subset of conceivable future projections. Our simulation results were thus contingent on the SSP output of AIM/Hub V2.2 and the associated extensions. Specifically, the long-term GDP growth assumed in this study may have been too optimistic for some SSP scenarios because they were simply extensions of existing SSPs. As a result, this study was more likely to provide an example of a long-term projection on a macroscopic scale than a reasonable vision that depicted prospective socioeconomic development using feasible technologies. Second, because of a lack of mitigation cost information for such a long-term projection, the MACs created using sensitivity data for times up to the year 2100 were extended to the year 2450. If the available technologies used beyond 2100 remained viable, this time-independent MAC may have been a plausible assumption. However, sophisticated technologies that are not currently available in our analysis may become viable and cost-competitive in the near future, and such technologies would be expected to significantly alter how energy is used in the future. In that case, the MACs we used would not apply to such advanced technologies for long-run projections.

In this study, we created a CB-IAM that combined a socioeconomic module and an RCM module to represent long-term projections for SSP1-5. We modified the socioeconomic module by adding a new SSP-dependent damage function, modeling SSPs using AIM/Hub V2.2 output, and extending the assessment duration to 2450. Our cost-benefit analysis indicated that all SSPs would exceed the 2 °C climate target by 2100. Except for SSP3, all SSPs would lead to a relatively moderate degree of temperature change (less than 1 °C in the long run). SSP5 would, in particular, stabilize at a near-zero temperature by the end of the evaluation period because of its comparatively low mitigation costs as well as the rapid growth of GDP, which made economies more willing to pay for the mitigation of future climate change losses. A cost-benefit analysis indicated that a high rate of GDP growth would contribute to mitigation of long-term climate change. In contrast, the SSP3 scenario led to a relatively substantial temperature increase over the entire evaluation period because the emissions were high, and it was challenging to reduce those emissions. The implication was that the Paris Agreement would be difficult to realize with such socioeconomic development, which would present significant obstacles to both mitigation and adaptation.

This study provided a valuable tool for undertaking a long-term cost-benefit analysis that considered distinct socioeconomic backgrounds. Furthermore, the study highlighted the need to examine the reduction of emissions from certain species while also taking into consideration various assumptions about socioeconomic development. The assumptions of the tool played a crucial role in identifying the optimal emission pathways. With this tool, we could generate estimates of socioeconomic and climate factors in the long run while taking into account the limitations imposed by climate policies. These policies included specific temperature targets, quantified carbon budgets, predefined emission paths, and other related constraints. In addition, the tool can be used to estimate the uncertainties that arise in various socioeconomic contexts and in the long-term climate system, both of which are crucial considerations for policymakers who formulate climate change strategies.



*Code availability.* The current version of the CB-IAM is available from the project website: https://github.com/sooxm/scm4eco. The exact version of the CB-IAM used to produce the results used in this paper is archived on Zenodo: https://doi.org/10.5281/zenodo.11928479.

*Data availability.* The sensitivity dataset for estimating carbon dioxide's marginal abatement cost (MAC) curves is available from Zenodo
(https://doi.org/10.5281/zenodo.11381113).

*Author contributions.* X.S., K. Takahashi, T.Y. and K. Tanaka designed the study. X.S. developed the model. S.F. provided the source code of AIM/Hub V2.2. J.T. provided the process-based impact simulation results. X.S. drafted the manuscript, with contributions from all other coauthors.

*Competing interests.* The authors declare that they have no competing financial interests.

*Acknowledgements.* This work was supported by the Decarbonized and Sustainable Society Research Program of the National Institute for Environmental Studies, Japan, and the Program for the advanced studies of climate change projection (SENTAN, Grant Number JP-MXD0722681344) from the Ministry of Education, Culture, Sports, Science and Technology (MEXT), Japan. X.S. received support from the Environment Research and Technology Development Fund (JPMEERF20202002) of the Environmental Restoration and Conservation Agency of Japan. K. Tanaka received support from the state assistance fund managed by the National Research Agency in France under the
Programme d'Investissements d'Avenir under the reference ANR-19-MPGA-0008.





## Appendix A: Model equations

### A1 Socioeconomic module

$$p_c(t, gas) = \theta_{1,gas}\mu(t,gas)^{\theta_{2,gas}} + \theta'_{1,gas}\mu(t,gas)^{\theta'_{2,gas}} \tag{A1}$$

$p_c(t, gas)$ Carbon price in year $t$ for emissions of $gas$.

$\mu(t, gas)$ Rate of control in year $t$ for emissions of $gas$.

$\theta_{1,gas}$, $\theta_{2,gas}$, $\theta'_{1,gas}$ and $\theta'_{2,gas}$ Estimated parameters for MAC curves (Table A1-A5).

$$\Delta_{abate,gas}(t) = \frac{1}{1000}Y_{gross}(t)\nu(t)\sigma(t,gas)\left(\frac{\theta_{1,gas}\mu(t,gas)^{\theta_{2,gas}+1}}{\theta_{2,gas}+1} + \frac{\theta'_{1,gas}\mu(t,gas)^{\theta'_{2,gas}+1}}{\theta'_{2,gas}+1}\right) \tag{A2}$$

$\Delta_{abate}(t, gas)$ Mitigation costs in trillions of USD (2010) in year $t$ for emissions of $gas$.

$\nu(t)$ Carbon price adjustment factor due to technological advancements in year $t$.

$$\Delta_{abate}(t) = \sum_{gas\in\{CO_2,CH_4,N_2O\}} \Delta_{abate}(t,gas) \tag{A3}$$

$\Delta_{abate}(t)$ Total mitigation costs in trillions of USD (2010) in year $t$.

$$\omega(t) = \omega(t_{2100}) \cdot \exp(-\frac{t-t_{2100}}{\tau_\omega}) \tag{A4}$$

$\omega(t)$ GDP per capita growth rate in year $t$.

$\omega(t_{2100})$ GDP per capita growth rate in year 2100.

$\tau_\omega$ Decay time, set as 50.

$$E_{abate}(t, gas) = \sigma(t,gas)Y_{gross}(t)\left(1-\mu(t,gas)\right) \tag{A5}$$

$\sigma(t, gas)$ Emission intensity per thousand USD (2010) in year $t$ for emissions of $gas$.

$E_{abate}(t, gas)$ Abatable emissions after abatement in year $t$ for emissions of $gas$.

$Y_{gross}(t)$ Gross output in trillions of USD (2010) in year $t$.





$\quad E_{non-abate}(t, gas) = E^{ref}_{non-abate}(t, gas)$ (A6)

$E_{non-abate}(t, gas)$ Non-abatable emissions in year $t$ for emissions of $gas$.

$E^{ref}_{non-abate}(t, gas)$ Reference non-abatable emissions in year $t$ for emissions of $gas$.

$$E(t, gas) = E_{abate}(t, gas) + E_{non-abate}(t, gas)$$ (A7)

$E(t, gas)$ Total emissions in year $t$ for emissions of $gas$.

$\quad E_{cca}(t+1) = E_{cca}(t) + E(t, CO_2)$ (A8)

$E_{cca}(t)$ Cumulative $CO_2$ emissions in year $t$.

$$Y_{gross}(t) = \frac{1}{1000} A(t) L(t)^{1-\gamma} K(t)^{\gamma}$$ (A9)

$A(t)$ Total factor productivity in year $t$.

$K(t)$ Capital in trillions of USD (2010) in year $t$.

$\quad L(t)$ Population/labor in millions in year $t$.

$\gamma$ Capital elasticity, $\gamma$=0.3.

$$\Delta_{gross}(t) = Y_{gross}(t) \left( a_1 T(t) + a_2 T(t)^{a_3} \right)$$ (A10)

$\Delta_{gross}(t)$ Gross climate damages in trillions of USD (2010) in year $t$.

$a_1$, $a_2$ and $a_3$ Tuned coefficients for climate change damages, see Table A8 for the estimated coefficients.

$\quad Y(t) = Y_{gross}(t) - \Delta_{abate}(t) - \Delta_{gross}(t)$ (A11)

$Y(t)$ Net production outputs in trillions of USD (2010) in year $t$.





$\Delta_{abate}(t)$ Total abatement costs in trillions of USD (2010) in year $t$.

$$C(t) = Y(t) - I(t) \tag{A12}$$

$C(t)$ Consumptions in trillions of USD (2010) in year $t$.

$I(t)$ Investment in trillions of USD (2010) in year $t$.

$$c(t) = 1000 \frac{C(t)}{L(t)} \tag{A13}$$

$c(t)$ Consumption per capita in thousands of USD (2010) in year $t$.

$$I(t) = S(t) \cdot Y(t) \tag{A14}$$

$S(t)$ Savings rates in year $t$.

$K(t+1) \leq K(t)(1 - \delta_K) + I(t) \tag{A15}$

$\delta_K$ Annual capital depreciation rate, which was taken to be 0.1.

$$U(t) = L(t) \cdot \log(c(t)) \tag{A16}$$

$U(t)$ Utility in year $t$.

$$W = \sum_{t=1850}^{t=2450} U(t) \cdot R(t) \tag{A17}$$

$W$ Total social welfare.

$R(t)$ Pure time preference discount factor in year $t$.

Here, the aggregate social welfare represents the combined utility from the years 1850 to 2450. The historical values specifically refer to the years 1850 to 2019, which remain constant. The optimization spanned the years 2020 to 2450.



## A2   Simple climate module

### A2.1   CO$_2$

$$\Delta RES_i(t) = \xi_{res,i} E(t, CO_2) - \frac{RES_i(t)}{\alpha_{res}(t)\tau_{res,i}} \tag{A18}$$

$RES_i(t)$ Carbon reservoir i (i = 1, 2, 3, 4) in year $t$ (GtC).

$\xi_{res,i}$ Fraction of CO$_2$ emissions entering reservoir i.

$\alpha_{res}(t)$ Scaling coefficient for carbon reservoirs.

$\tau_{res,i}$ Time coefficient for reservoir i (years).

$$C_{atm}(t) - C_{atm}^0 = \sum_{i=1}^{4} RES_i(t) \tag{A19}$$

$C_{atm}^0$ Carbon in the atmospheric pool (GtC) in year 1850.

$$C_{acc}(t) = \sum_{i=1850}^{t} E(t, CO_2) - (C_{atm}(t) - C_{atm}^0) \tag{A20}$$

$C_{acc}(t)$ Accumulated carbon stock in the carbon pools other than atmosphere (land and ocean) in year $t$ (GtC).

$$IRF_{100}(t) = \zeta_0 + \zeta_C C_{acc}(t) + \zeta_T T_{atm}(t) \tag{A21}$$

$IRF_{100}(t)$ 100-year integrated impulse-response function.

$\zeta_0, \zeta_C$ and $\zeta_T$ Coefficients of the impulse-response function.

$T_{atm}(t)$ Global mean atmospheric temperature in year $t$.

$$IRF_{100}(t) = \sum_{i=1}^{4} \alpha_{res}(t)\xi_{res,i}\tau_{res,i} \left[1 - exp(\frac{-100}{\alpha_{res}(t)\tau_{res,i}})\right] \tag{A22}$$

$$C_{CO_2}(t) = \frac{C_{atm}(t)}{\alpha_{ppm2gtc}} \tag{A23}$$





$C_{atm}(t)$ Carbon in the atmospheric pool (GtC) in year $t$.

$C_{CO_2}(t)$ CO$_2$ concentration in (ppm) in year $t$.

$\alpha_{ppm2gtc}$ Unit conversion factor from ppm to GtC, taken to be 2.123 GtC ppm$^{-1}$.

$$f_{CO_2}(t) = \alpha_{CO_2} \log \frac{C_{CO_2}(t)}{C_{CO_2}^0} \tag{A24}$$

$f_{CO_2}(t)$ CO$_2$ radiative forcing in year $t$.

$\alpha_{CO_2}$ Scaling parameter, assumed to be $\frac{3.71}{\log(2)}$=5.35 Wm$^{-2}$.

$C_{CO_2}^0$ CO$_2$ concentration of preindustrial level, taken to be 278 ppm.

### A2.2    CH$_4$ & N$_2$O

$$\Delta C_{CH_4}(t) = \frac{E(t,CH_4)}{\theta_{CH_4}} - \frac{C_{CH_4}(t-1)}{\tau_{CH_4}^{tot}(t-1)} \tag{A25}$$

$\Delta C_{CH_4}(t)$ CH$_4$ concentration (ppb) in year $t$.

$E(t,CH_4)$ CH$_4$ emissions (MtCH$_4$ yr$^{-1}$) in year $t$.

$\theta_{CH_4}$ conversion factor, which is taken to be 2.78 Tg ppb$^{-1}$.

$\tau_{CH_4}^{tot}(t-1)$ Total lifetime of CH$_4$.

$$\frac{1}{\tau_{CH_4}^{tot}(t)} = \frac{1}{\tau_{CH_4}^{init}/\tau_{OH}^{rel}(t)} + \frac{1}{\tau_{CH_4}^{soil}} + \frac{1}{\tau_{CH_4}^{oth}} \tag{A26}$$

$\tau_{CH_4}^{init}/\tau_{OH}^{rel}(t)$ Lifetimes of CH$_4$ in troposphere.

$\tau_{CH_4}^{soil}$ Lifetimes of CH$_4$ in soil, taken to be 160 years.

$\tau_{CH_4}^{oth}$ Lifetimes of CH$_4$ in stratosphere, taken to be 120 years.

$$\begin{aligned}
\tau_{OH}^{rel}(t) = \ & S_{\tau_{CH_4}} \Delta T_{2k}(t) \\
& + \left( \frac{C_{CH_4}(t)}{C_{CH_4}^{2k}} \right)^{S_{CH_4}^{OH}} \exp\left( S_{NO_x}^{OH} \Delta E_{NO_x}(t) + S_{CO}^{OH} \Delta E_{CO}(t) + S_{VOC}^{OH} \Delta E_{VOC}(t) \right)
\end{aligned} \tag{A27}$$



$S_x^{OH}$ Sensitivities of the tropospheric OH to CH$_4$, NO$_x$, CO and VOC, with values of -0.32, +0.0042, -1.05E-4 and -3.15E-4, respectively.

$C_{CH_4}^{2k}$ CH$_4$ concentration (ppb) in the year 2000.

$S_{\tau_{CH_4}}$ Temperature sensitivity coefficient, taken to be 0.0316 °C$^{-1}$.

$\Delta T_{2k}(t)$ Change in the temperature above the 2000 level.

$$\tau_{N_2O}(t) = \tau_{N_2O}^{init} \left( \frac{C_{N_2O}(t)}{C_{N_2O}^{2k}} \right)^{S_{\tau_{N_2O}}} \tag{A28}$$

$C_{N_2O}(t)$ N$_2$O concentration (ppb) in year $t$.

$\tau_{N_2O}(t)$ N$_2$O lifetime (years).

$\tau_{N_2O}^{init}$ Initial lifetime of N$_2$O, taken to be 120 years.

$C_{N_2O}^{2k}$ N$_2$O concentration in the year 2000.

$S_{\tau_{N_2O}}$ N$_2$O sensitivity coefficient, taken to be -0.05.

$$\Delta C_{N_2O}(t) = \frac{E(t, N_2O)}{\theta_{N_2O}} - \frac{C_{N_2O}(t-1)}{\tau_{N_2O}(t-1)} \tag{A29}$$

$\Delta C_{N_2O}(t)$ Change in the atmospheric N$_2$O concentration (ppb).

$E(t, N_2O)$ N yr$^{-1}$ emissions (MtN$_2$O-N yr$^{-1}$).

$\theta_{N_2O}$ Conversion factor for N$_2$O, taken to be 4.81 Tg ppb$^{-1}$.

$$f_{CH_4}(t) = \alpha_{CH_4} \left( \sqrt{C_{CH_4}(t)} - \sqrt{C_{CH_4}^0} \right) - \left( f_{mn} \left( C_{CH_4}(t), C_{N_2O}^0 \right) - f_{mn} \left( C_{CH_4}^0, C_{N_2O}^0 \right) \right) \tag{A30}$$

$$f_{N_2O}(t) = \alpha_{N_2O} \left( \sqrt{C_{N_2O}(t)} - \sqrt{C_{N_2O}^0} \right) - \left( f_{mn} \left( C_{CH_4}^0, C_{N_2O}(t) \right) - f_{mn} \left( C_{CH_4}^0, C_{N_2O}^0 \right) \right) \tag{A31}$$

$f_{CH_4}(t)$ Radiative forcings of CH$_4$ ($Wm^{-2}$).

$f_{N_2O}(t)$ Radiative forcings of N$_2$O ($Wm^{-2}$).





$\alpha_{CH_4}$ Scaling factors for CH$_4$, taken to be 0.036.

$\alpha_{N_2O}$ Scaling factors for N$_2$O, taken to be 0.12.

$C_{CH_4}^0$ Pre-industrial concentrations of CH$_4$, taken to be 721.9 ppb.

$C_{N_2O}^0$ Pre-industrial concentrations of N$_2$O, taken to be 273.0 ppb.

$$f_{mn}(M,N) = 0.47 \log \left( 1 + 0.6356 \left( \frac{MN}{10^6} \right)^{0.75} + 0.007 \frac{M}{10^3} \left( \frac{MN}{10^6} \right)^{1.52} \right) \tag{A32}$$

$f_{mn}(M,N)$ Overlap between CH$_4$ and N$_2$O, $M$ and $N$ are the CH$_4$ and N$_2$O concentration inputs, respectively.

**A2.3 Halogenated gases**

$$C_{hc}(t+1,hc) = \tau_{hc} \frac{E(t,hc)}{\mu_{hc}} \frac{\rho_{atm}}{m_{atm}} \left( 1 - \exp(-\frac{1}{\tau_{hc}}) \right) + C_{hc}(t,hc) \left( 1 - \exp(-\frac{1}{\tau_{hc}}) \right) \tag{A33}$$

$C_{hc}(t+1,hc)$ Concentration (in ppt) of halogenated gas $hc$ in year $t+1$.

$E(t,hc)$ Emission level of $hc$ in kt yr$^{-1}$.

$\mu_{hc}$ Molar mass of $hc$.

$\tau_{hc}$ Lifetime of $hc$.

$\rho_{atm}$ Average density of air.

$m_{atm}$ Total mass of the atmosphere.

$$f_{hc}(t,hc) = \alpha_{hc} \left( C_{hc}(t,hc) - C_{hc}^0 \right) \tag{A34}$$

$f_{hc}(t,hc)$ Radiative forcing from halogenated gas $hc$ in year t.

$\alpha_{hc}$ Radiative efficiency.

$C_{hc}^0$ Pre-industrial atmospheric concentration for halogenated gas $hc$.





### A2.4   Direct effect of aerosols

$$f_{dir}^{ind}(t, aero) = \alpha_{dir}^{ind}(aero) \frac{E^{ind}(t, aero)}{E_0^{ind}(aero)} \tag{A35}$$

$$f_{dir}^{lnd}(t, aero) = \alpha_{dir}^{lnd}(aero) \frac{E^{lnd}(t, aero)}{E_0^{lnd}(aero)} \tag{A36}$$

$f_{dir}^{ind}(t, aero)$ Radiative forcings from industrial aerosols ($aero \in \{SO_x, NOx, BC, OC\}$).

$f_{dir}^{lnd}(t, aero)$ Radiative forcings from land-use aerosols $aero \in \{SO_x, NOx, BC, OC\}$.

$\alpha_{dir}^{ind}(aero)$ Industrial aerosol forcing levels in 2005 ($Wm^{-2}$).

$\alpha_{dir}^{lnd}(aero)$ Land-use aerosol forcing levels in 2005 ($Wm^{-2}$).

$E^{ind}(t, aero)$ and $E^{lnd}(t, aero)$ Industrial and land-use aerosol emissions in year $t$.

$E_0^{ind}(aero)$ and $E_0^{lnd}(aero)$ Industrial and land-use aerosol emissions in the reference year of 2005.

$$f_{bio}(t) = \sum_{aero} f_{dir}^{lnd}(t, aero) \tag{A37}$$

$f_{bio}(t)$ Biomass aerosol forcing in year $t$.

### A2.5   Mineral dust aerosols

$$f_{mindust}(t) = -0.1 \tag{A38}$$

$f_{mindust}(t)$ Radiative forcing from mineral dust aerosols.

### A2.6   Tropospheric aerosols

$$f_{alb}(t) = rP_{alb}(t) \log \left( \frac{\sum_{aero} \omega_{aero} N_{aero}(t)}{\sum_{aero} \omega_{aero} N_{aero}^0} \right) \tag{A39}$$

$f_{alb}(t)$ "First" type of indirect forcing.

$P_{alb}(t)$ Indirect aerosol effects related to the albedo.

$r$ Aerosol scaling factor, taken to be 0.5383.

$\sum_{aero} \omega_{aero} N_{aero}(t)$ Total aerosol number concentration.

$\sum_{aero} \omega_{aero} N_{aero}^0$ Pre-industrial level of total aerosol number concentration.





$N_{aero}$ Number concentrations from aerosol masses.

$\omega_{aero}$ Aerosol contribution shares.

$$f_{cov}(t) = 0 \tag{A40}$$

$f_{cov}(t)$ Second indirect effect on the cloud cover changes.

### A2.7 Stratospheric ozone

$$C_{EESC}(t) = a_{EESC} \left( \sum_{Cl} n_{Cl} f_{Cl} C_{hc}(t, Cl) + \alpha_{br} \sum_{Br} n_{Br} f_{Br} C_{hc}(t, Br) \right) \tag{A41}$$

$C_{EESC}(t)$ Equivalent effective stratospheric chlorine concentration (EESC, in ppb).

$n_{Cl}$ and $n_{Br}$ Numbers of chlorine and bromine atoms, respectively.

$f_{Cl}$ and $f_{Br}$ Release efficiencies of stratospheric halogens.

$C_{hc}(t, Cl)$ and $C_{hc}(t, Br)$ Gas mixing rates in the stratosphere.

$\alpha_{br}$ Ratio of effectiveness in ozone depletion between bromine and chlorine, assumed to be 45.

$a_{EESC}$ Fractional release factor, taken to be 0.7711.

$$f_{strat}(t) = \eta_1 \left( \eta_2 \Delta C_{EESC}(t) \right)^{\eta_3} \tag{A42}$$

$f_{strat}(t)$ Forcing effect of the depletion of stratospheric ozone.

$\Delta C_{EESC}(t)$ $C_{EESC}(t)$ concentration above the 1980 level.

$\eta_1$, $\eta_2$ and $\eta_3$ $\eta_1$ is the sensitivity scaling factor, calibrated to be -0.00106; $\eta_2 = \frac{1}{100}$ ppb$^{-1}$; and $\eta_3$ is the sensitivity exponent, taken to be 1.7.

### A2.8 Tropospheric ozone

$$C_{troz}(t) = \begin{cases} C_{troz}^{his}(t), & \text{if } t < 2000 \\ C_{troz}^{2k} + \omega_{CH_4} \log \left( \dfrac{C_{CH_4}(t)}{C_{CH_4}^{2k}} \right) + \omega_{NO_x} \left( E_{NO_x}(t) - E_{NO_x}^{2k} \right) \\ \qquad + \omega_{CO} \left( E_{CO}(t) - E_{CO}^{2k} \right) + \omega_{VOC} \left( E_{VOC}(t) - E_{VOC}^{2k} \right), & \text{if } t \geq 2000 \end{cases} \tag{A43}$$



$C_{troz}^{his}(t)$ Historical tropospheric ozone concentrations, taken directly from the output of MAGICC 6.0.

$C_{troz}(t)$ Tropospheric ozone concentrations.

$C_{troz}^{2k}$ Tropospheric ozone concentration in the year 2000.

$\omega_j$ ($j \in \{CH_4, NO_x, CO, VOC\}$) Coefficients describing the sensitivity of the tropospheric ozone to CH$_4$, NO$_x$, CO and VOC, which are taken to be 5.0, 0.125, 0.0011 and 0.0033, respectively.

$$f_{troz}(t) = \alpha_{troz}\left(C_{troz}(t) - C_{troz}^0\right) \tag{A44}$$

$f_{troz}(t)$ Radiative forcing from tropospheric ozone.

$C_{troz}^0$ pre-industrial concentration of tropospheric ozone taken to be 19.9481 DU.

$\alpha_{troz}$ Radiative efficiency factor of tropospheric ozone, taken to be 0.0335 Wm$^{-2}$ DU$^{-1}$.

### A2.9    Stratospheric water vapor from CH$_4$ oxidation

$$f_{H_2O}(t) = \alpha_{H_2O}\left(\alpha_{CH_4}\left(\sqrt{C_{CH_4}(t)} - \sqrt{C_{CH_4}^0}\right)\right) \tag{A45}$$

$f_{H_2O}(t)$ Forcing effect of stratospheric water vapor from CH$_4$ oxidation.

$\alpha_{CH_4}$ Forcing efficiency of stratospheric water vapor from CH$_4$ oxidation, taken to be 0.036

$\alpha_{H_2O}$ Fraction factor related to CH$_4$ forcing, which is taken to be 15%.

### A2.10    Land-use albedo

$$f_{landuse}(t) = -0.2 \tag{A46}$$

$f_{landuse}(t)$ Forcing effect from the land-use albedo.

### A2.11    BC on snow

$$f_{BCSnow}(t) = a_{BC} + E(t, BC)b_{BC} \tag{A47}$$

$f_{BCSnow}(t)$ Forcing effect of the BC on snow.

$E(t, BC)$ BC emissions.

$a_{BC}$ and $b_{BC}$ Estimated parameters, taken to be -0.005265 and 0.01277, respectively.





### A2.12 Natural sources

$$f_{volc}(t) = f_{volc}^{CMIP6}(t) \tag{A48}$$

$$f_{solar}(t) = f_{solar}^{CMIP6}(t) \tag{A49}$$

$f_{volc}(t)$ Volcanic forcings.

$f_{solar}(t)$ Solar forcings.

$f_{volc}^{CMIP6}(t)$ Volcanic forcings of CMIP6.

$f_{solar}^{CMIP6}(t)$ Solar forcings of CMIP6.

### A2.13 Effective radiative forcing

$$f_x^e(t) = E_x^a \cdot f_x(t) \tag{A50}$$

$$f^e(t) = \sum_x^X f_x^e(t) \tag{A51}$$

$f_x(t)$ Radiative forcing of source $x$ $(W m^{-2})$.

$E_x^a$ Forcing efficacy of source $x$ $(W m^{-2})$.

$f_x^e(t)$ Effective radiative forcing of source $x$ $(W m^{-2})$.

$f^e(t)$ Total effective radiative forcing $(W m^{-2})$.





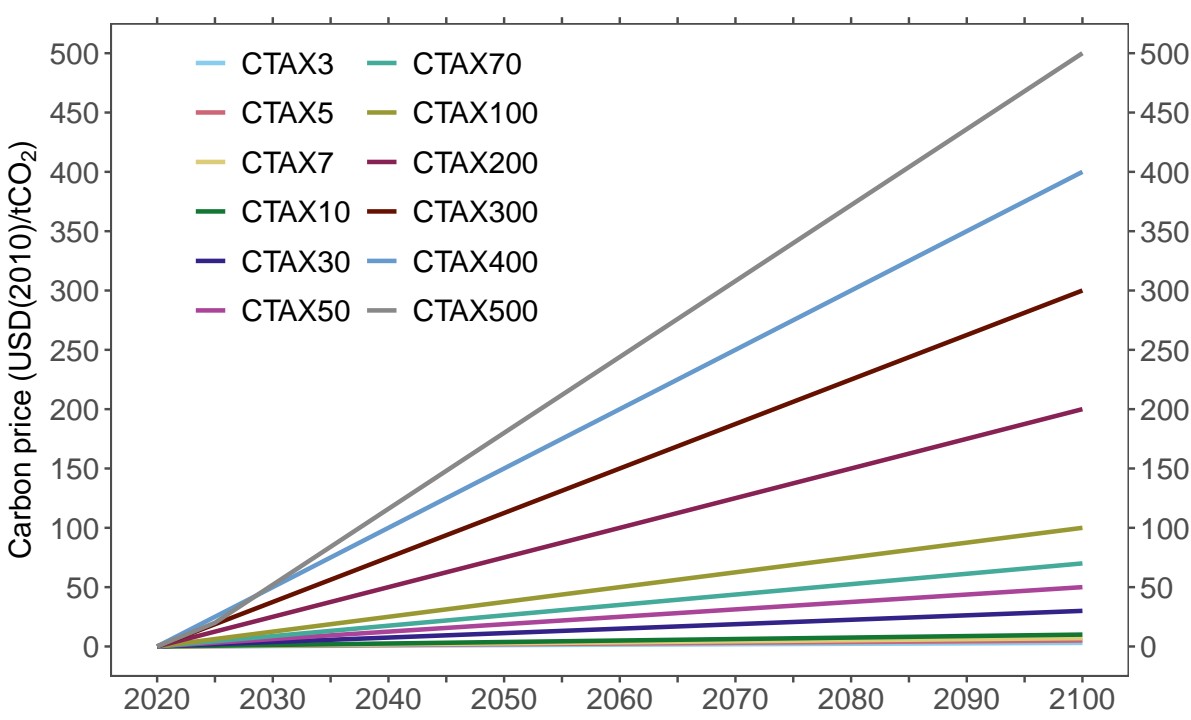

**Figure A1. Carbon price paths used to generate the sensitivity dataset.** Colored lines show the carbon price paths from 2020 to 2100.



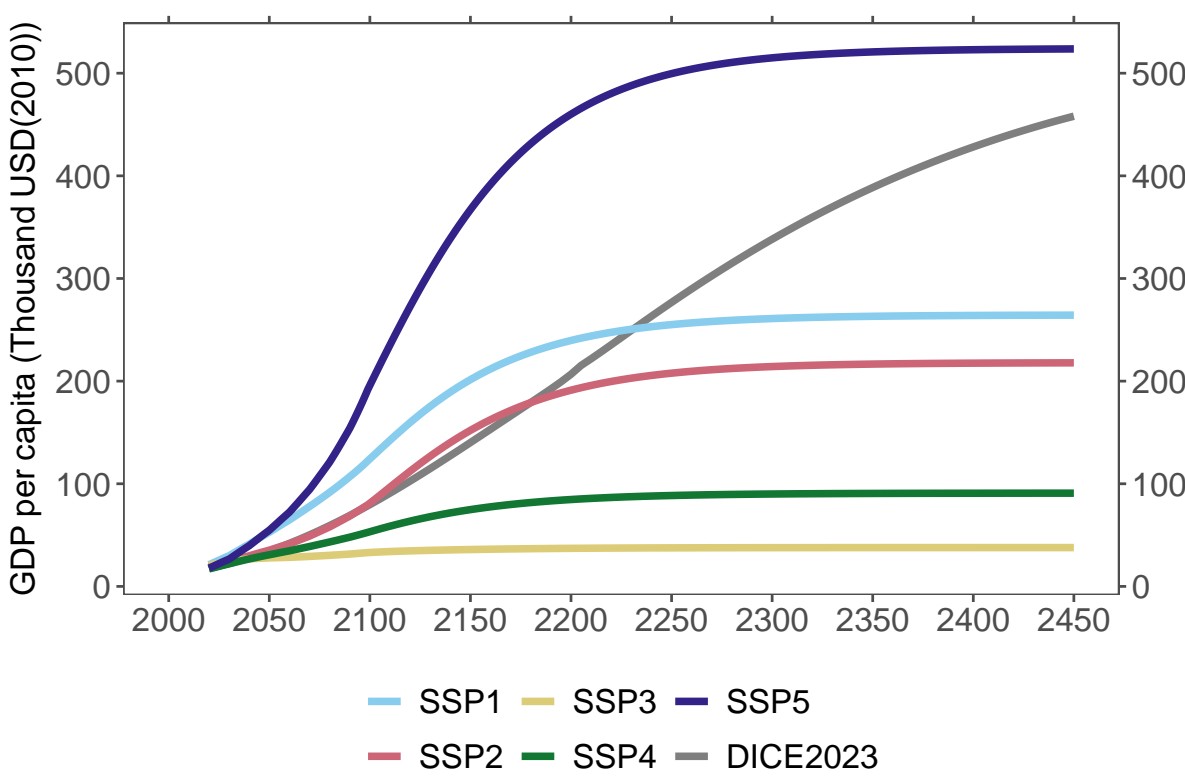

**Figure A2. GDP per capita for reference scenario.** Colored lines show the SSP projections, and the gray line indicates the assumptions of DICE2023.



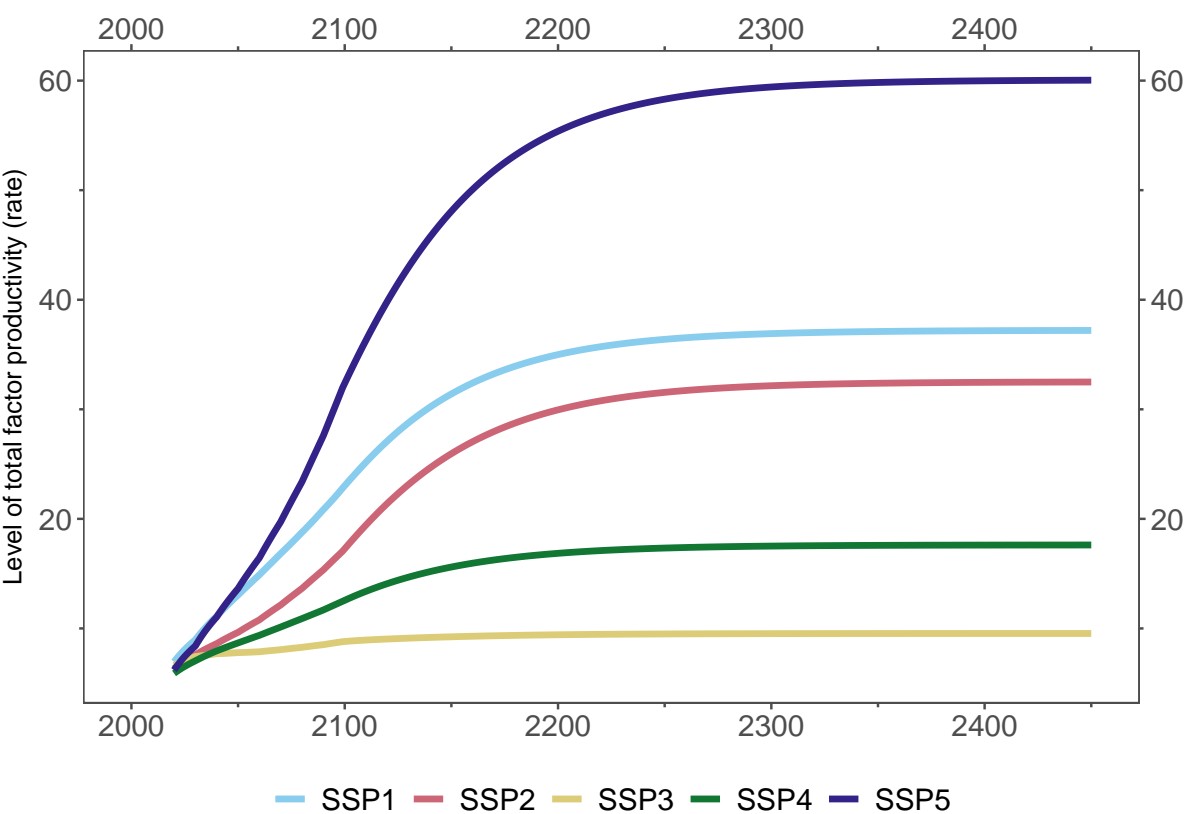

**Figure A3. Total factor productivity.** Colored lines show the SSP projections.



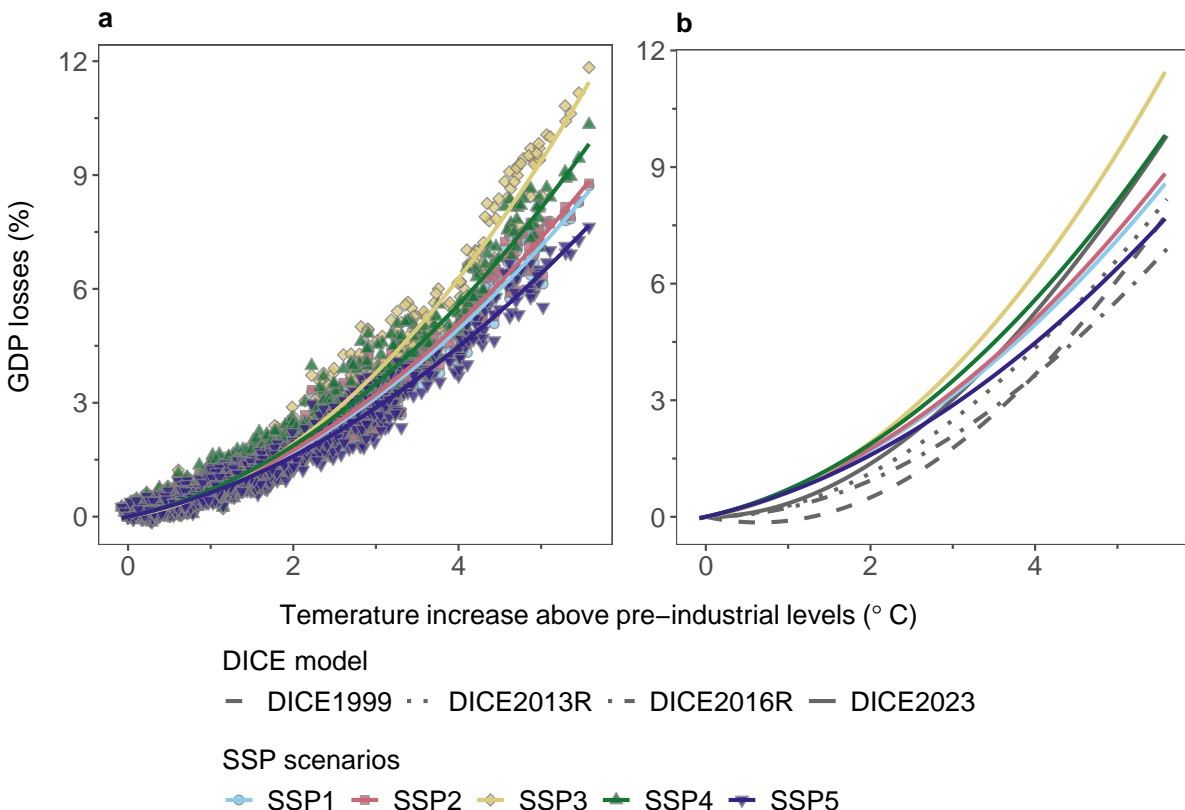

**Figure A4. Damage functions in the DICE model and this study. a** Sensitivity data derived from the process-based impact simulation results (Takakura et al., 2021), and the estimated SSP-dependent damage functions. Colored points show the relationship between GDP losses and temperature increases above pre-industrial levels. Colored lines indicate the estimated damage functions. **b** The estimated SSP-dependent damage functions in relation to various versions of the DICE model. Colored lines denote the estimated damage functions, and the gray lines show the MAC curves used in the DICE model. See Table A8 for the estimated coefficients.





**Figure A5. Rate of control of emissions subject to the constraints of carbon price pathways for the SSP1 scenario.** Colored points show the rates of control of emissions derived from AIM/Hub V2.2. Colored lines show the paths calculated using the CB-IAM.





**Figure A6. Rate of control of emissions subject to the constraints of carbon price pathways for the SSP2 scenario.** Colored points show the rates of control of emissions derived from AIM/Hub V2.2. Colored lines show the paths calculated using the CB-IAM.







**Figure A7. Rate of control of emissions subject to the constraints of carbon price pathways for the SSP3 scenario.** Colored points show the rates of control of emissions derived from AIM/Hub V2.2. Colored lines show the paths calculated using the CB-IAM.





**Figure A8. Rate of control of emissions subject to the constraints of carbon price pathways for the SSP4 scenario.** Colored points show the rates of control of emissions derived from AIM/Hub V2.2. Colored lines show the paths calculated using the CB-IAM.





**Figure A9. Rate of control of emissions subject to the constraints of carbon price pathways for the SSP5 scenario.** Colored points show the rates of control of emissions derived from AIM/Hub V2.2. Colored lines show the paths calculated using the CB-IAM.



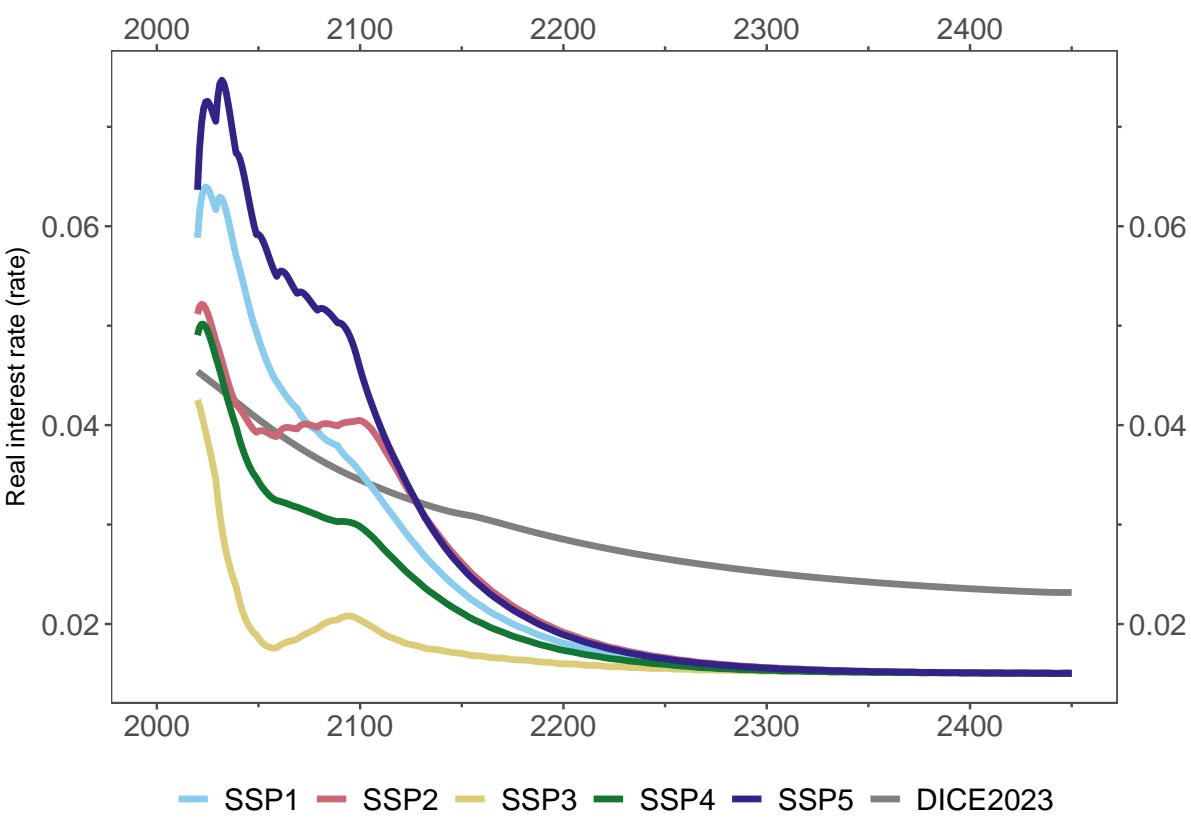

**Figure A10. Discount rates for the SSPs.** Colored lines show the SSP projections, and the gray line indicates the assumption of DICE2023.



**Table A1.** The estimated parameters for SSP1's MAC curves.

|  | $\theta_{1,gas}$ | $\theta'_{1,gas}$ | $\theta_{2,gas}$ | $\theta'_{2,gas}$ |
|---|---|---|---|---|
| $CO_2$ | 156.2 | 1.0 | 2.3 | 14.1 |
| $CH_4$ | 240.2 | 3726.4 | 4.2 | 45.6 |
| $N_2O$ | 122.6 | 2156.7 | 3.3 | 24.9 |
| $SO_x$ | 219.4 | 55933.5 | 2.1 | 76.1 |
| CO | 211.3 | 4838.4 | 1.5 | 17.9 |
| VOC | 248.4 | 6840.6 | 2.2 | 24.5 |
| $NO_x$ | 212.8 | 4841.0 | 1.8 | 30.4 |
| BC | 227.5 | 5973.7 | 1.8 | 26.5 |
| OC | 230.4 | 1825.6 | 2.7 | 34.1 |



**Table A2.** The estimated parameters for SSP2's MAC curves.

|  | $\theta_{1,gas}$ | $\theta'_{1,gas}$ | $\theta_{2,gas}$ | $\theta'_{2,gas}$ |
|---|---|---|---|---|
| $CO_2$ | 192.2 | 1.0 | 1.6 | 13.8 |
| $CH_4$ | 708.9 | 1000000.0 | 5.3 | 64.4 |
| $N_2O$ | 148.9 | 2327.2 | 2.7 | 13.2 |
| $SO_x$ | 315.4 | 6170.5 | 2.2 | 26.2 |
| $CO$ | 580.2 | 2425.6 | 2.9 | 13.6 |
| $VOC$ | 520.7 | 16949.4 | 2.5 | 27.1 |
| $NO_x$ | 423.2 | 7153.4 | 1.7 | 14.4 |
| $BC$ | 772.4 | 224196.0 | 3.5 | 36.5 |
| $OC$ | 344.6 | 1674.5 | 1.9 | 10.5 |





**Table A3.** The estimated parameters for SSP3's MAC curves.

|  | $\theta_{1,gas}$ | $\theta'_{1,gas}$ | $\theta_{2,gas}$ | $\theta'_{2,gas}$ |
|---|---|---|---|---|
| $CO_2$ | 235.7 | 99.6 | 1.2 | 10.0 |
| $CH_4$ | 600.3 | 39104.4 | 4.4 | 27.0 |
| $N_2O$ | 204.9 | 4177.9 | 2.4 | 10.4 |
| $SO_x$ | 236.3 | 2288.0 | 2.0 | 15.3 |
| CO | 501.1 | 3432.1 | 2.5 | 11.1 |
| VOC | 511.7 | 2493.5 | 2.6 | 14.5 |
| $NO_x$ | 365.7 | 22950.8 | 1.3 | 9.4 |
| BC | 708.2 | 37398.1 | 3.1 | 18.8 |
| OC | 213.7 | 2581.8 | 1.0 | 9.2 |





**Table A4.** The estimated parameters for SSP4's MAC curves.

|  | $\theta_{1,gas}$ | $\theta'_{1,gas}$ | $\theta_{2,gas}$ | $\theta'_{2,gas}$ |
|---|---|---|---|---|
| $CO_2$ | 124.4 | 0.0002 | 1.2 | 20.0 |
| $CH_4$ | 247.5 | 4564.5 | 4.3 | 41.2 |
| $N_2O$ | 120.3 | 2182.9 | 3.1 | 20.7 |
| $SO_x$ | 191.2 | 5373.7 | 2.3 | 30.3 |
| CO | 267.8 | 10807.3 | 1.8 | 18.5 |
| VOC | 451.0 | 64889.8 | 3.1 | 47.3 |
| $NO_x$ | 223.0 | 12332.9 | 1.7 | 28.4 |
| BC | 231.6 | 4471.1 | 1.8 | 21.8 |
| OC | 167.8 | 828.2 | 2.2 | 29.9 |





**Table A5.** The estimated parameters for SSP5's MAC curves.

|     | $\theta_{1,gas}$ | $\theta'_{1,gas}$ | $\theta_{2,gas}$ | $\theta'_{2,gas}$ |
|-----|--------|--------|--------|--------|
| $CO_2$ | 157.8 | 0.006 | 1.2 | 20.0 |
| $CH_4$ | 24.3 | 782.5 | 1.0 | 5.1 |
| $N_2O$ | 73.4 | 1780.4 | 1.5 | 7.0 |
| $SO_x$ | 167.3 | 983.2 | 1.1 | 11.0 |
| CO | 410.6 | 461.1 | 1.4 | 4.4 |
| VOC | 387.2 | 272.1 | 1.5 | 4.8 |
| $NO_x$ | 202.8 | 879.5 | 1.0 | 4.8 |
| BC | 320.5 | 540.3 | 1.5 | 4.5 |
| OC | 200.8 | 996.0 | 1.0 | 5.9 |





**Table A6.** List of sectors within which mitigations of emissions were considered in this study.

| Sectors | $CO_2$ | $CH_4$ | $N_2O$ | Halogenated gases | BC | CO | $NO_x$ | $SO_x$ | OC | VOC |
|---|---|---|---|---|---|---|---|---|---|---|
| AFOLU | × | × | × | - | × | × | × | × | × | × |
| Energy Demand for AFOFI | √ | - | - | - | - | - | - | - | - | - |
| Energy Demand for Commercial | √ | √ | √ | - | × | × | × | × | × | × |
| Energy Demand for Industry | √ | √ | - | - | × | × | × | × | × | × |
| Energy Demand for Residential | √ | √ | √ | - | × | × | × | × | × | × |
| Energy Demand for Transportation | √ | √ | √ | - | √ | √ | √ | √ | √ | √ |
| Energy Supply | √ | √ | - | - | √ | √ | √ | √ | √ | √ |
| Industrial Processes | √ | √ | √ | × | × | × | × | × | × | × |
| Waste | √ | √ | √ | - | × | × | × | × | × | × |

Table note

√: Included; ×: Not included; -: Not available.

AFOLU: Agriculture, Forestry and Other Land Use; AFOFI: Energy Demand for Agriculture, Forestry and Fishing.



**Table A7.** Main variables considered for the extended SSP reference scenarios

| Variables | Historical periods (1850-2019) | SSP projections (2020-2100) | SSP extensions (2101-2450) |
|---|---|---|---|
| Population | Obtained from a UN report (United Nations Secretariat, 1999) and the UN Population Databases (*) | AIM/Hub V2.2 output | Extended in this study, see section 2.2.1 |
| Gross output | Assumed to follow (Maddison, 2007)'s estimation, AIM/Hub V2.2 output | AIM/Hub V2.2 output | Extended in this study, see section 2.2.1 |
| Investment | Estimated based on Eq. A14 | Estimated based on Eq. A14 | Estimated based on Eq. A14 |
| Consumption | Estimated based on Eq. A12 | Estimated based on Eq. A12 | Estimated based on Eq. A12 |
| Capital stock | Estimated based on Eq. A15 | Estimated based on Eq. A15 | Estimated based on Eq. A15 |
| Abatable GHGs | CEDS (Hoesly et al., 2018), EDGAR v7.0_GHG 1970-2021 and EDGAR v6.1_AP 1970-2018 (Crippa et al., 2021, 2022) | AIM/Hub V2.2 output | Extended in this study, see section 2.2.2 |
| Non-abatable GHGs | CEDS (Hoesly et al., 2018), EDGAR v7.0_GHG 1970-2021 and EDGAR v6.1_AP 1970-2018 (Crippa et al., 2021, 2022) | AIM/Hub V2.2 output | Extended in this study, see section 2.2.2 |
| Abatable aerosols | CEDS (Hoesly et al., 2018), EDGAR v7.0_GHG 1970-2021 and EDGAR v6.1_AP 1970-2018 (Crippa et al., 2021, 2022) | AIM/Hub V2.2 output | Extended in this study, see section 2.2.2 |
| Non-abatable aerosols | CEDS (Hoesly et al., 2018), EDGAR v7.0_GHG 1970-2021 and EDGAR v6.1_AP 1970-2018 (Crippa et al., 2021, 2022) | AIM/Hub V2.2 output | Extended in this study, see section 2.2.2 |

(*) UN Population Databases: https://population.un.org/wpp/





**Table A8.** Estimated coefficients for climate damage functions

| Symbols | $a_1$ | $a_2$ | $a_3$ |
|---|---|---|---|
| Names | Damage intercept | Damage quadratic term | Damage exponent |
| SSP1 | 0.00476 | 0.00190 | 2 |
| SSP2 | 0.00480 | 0.00198 | 2 |
| SSP3 | 0.00347 | 0.00306 | 2 |
| SSP4 | 0.00476 | 0.00230 | 2 |
| SSP5 | 0.00471 | 0.00162 | 2 |





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
