# Peer review of "Developing an integrated assessment model to explore optimal cost-benefit paths for Shared Socioeconomic Pathways scenarios"

_EGUsphere, 2024_

## Referee Comment (RC2)

**Notes on Developing an integrated assessment model to explore optimal cost-benefit paths for Shared Socioeconomic Pathways scenarios**

**General comments:**

This paper attempts to join the extensive field of scenario-generating IAMs with the field of cost-benefit analysis IAMS. Some of the updates it proposes are valuable, however more work will be required to ensure that the progress in joining the fields is rigorous. It frequently does not justify model assumptions, and some of its assumptions, particularly around time-dependence, do not seem easy to justify.

**Specific comments:**

The paper largely frames itself as a response to DICE. While DICE is certainly the best-known CBA model, it is hardly the only one, and other families (those based on PAGE, for instance [1-2]) should also be cited. Recognising the diversity of existing CBA models will also highlight the importance of giving your model a real name – "the CBA-IAM" is not precise.

One of the major differences between SSPs is regional development. AIM includes regional information but It's completely unclear whether and how you use this in your calculations. Definitions of what counts as the "optimal scenario" are strongly dependent on concepts of inequality and justice, which are so thoroughly unexplored in this work that I have to read the appendix to know how you define the titular word "optimal". This must be clarified, justified and preferably diversified in the analysis. You use a discounting function with an elasticity of marginal utility over time, one would imagine the same constant should also apply over space when considering the damage function.

Equation 1 presents some problems. The points in plots A5-A10 should lie on the lines, since you're entirely fitting the relationship to the trends in AIM, and they don't fit at all. I assume the problem is that the MAC curve is invariant over time, but learning effects mean that emissions reductions get much cheaper with time. By compressing all time into the one plot, you obscure the temporal effects in systematically biasing ways, since higher $\mu$ tend to occur at later times. Trying to model centuries of economic development without a detailed investigation into learning rates seems crazy. This can be fixed with a time-dependent relationship.

The equation also obscures the link between the carbon price and emissions across species – possibly this is a writeup problem rather than a code problem. Normally there is an assumption that mitigation across species is linked both politically and through technology, but here you appear able to control each species separately via a species-specific emissions price. I don't believe your damage function includes non-thermal damages from smog, so you can't allow these to be entirely free (or you'd dial up smog for coolness). Do you post-hoc require the species to all have the same "carbon" price except when testing carbon-only mitigation?

The sheer density of equations used in the paper is impressive, but for every equation I need to have some confidence in every term used, or at least know that you have done some level of analysis to show that the model is robust against it being substantially misestimated. In practice, almost none of the values have easily-found citations or justifications, and I'm sure some of them (e.g. the 50-year GDP decay in the extension, with no underlying economic reason

given for a decay in the first place) will have strong impacts on the projections. You could assuage concern over the climate model by comparing the reconstructed and observed temperature rise.

**Technical corrections:**

5: "the most recent integrated assessment model": which one? I doubt this is still the newest even now.

11. In the main text, you seem (justifiably) dismissive of the value of making multi-century projections of GDP/climate damages and put little effort into doing this in the paper. Other papers already do this to a similar extent, so including it as a key motive behind the paper in the abstract seems odd.

Intro: Since many aspects of the paper revolve around detailed discussions of what the SSP pathways represent, it would probably be helpful to have a short overview of what they are.

30. This extremely narrow range of "baseline" temperature values says a lot about the range of papers you are citing.

60. SSPs do not attempt to "forsee" the future, they merely create narratives about what it may involve.

75: the way the MAC curve calculations are discussed isn't very clear, I'd recommend rewriting it starting with the general principles used to calculate individual trajectories in AIM, then proceeding to the details of how the specific trajectories were discovered. It certainly needs a lot more details.

77. Linear growth in carbon prices is not generally economically optimal, and the fact that the lead author has done it before is not a good justification for doing it again. It seems unlikely you will create "optimal" paths to net zero by interpolation between nonoptimized pathways over time. However, the plot in the SI shows that the CTAX500 pathway is actually bilinear, with a different gradient between 2020 and 2025 compared to thereafter. Is this a mistake? Or are you trying to limit the historical error in this path?

81: By "total emissions", do you mean emissions in the world with zero carbon price? Please define this key concept more clearly.

93: The low sensitivity of land use emissions to carbon price is not unexpected, though it doesn't really justify leaving the emissions out entirely, since there is an effect and it doesn't seem hard to include all sectors in AIM in your model.

170: Where has *v(t)* been all this time? What determines it? This isn't explained either here or in the appendix.

180: DICE does not fall within your estimates at small T.

213: This problem seems pretty damning but entirely solvable by using pathway-dependent or time-dependent MACCs. Given the discount rate and lock-in effects (which you are ignoring here), the near term has a larger impact on optimal prices than the long term.

Figure 7: Axis markings on the inner graphs would make this more readable, or at least a line at control rate = 1.

285: "Defaulted to"? You set it to this value, surely. This is one important aspect of climate-related uncertainty, but the rate of adjustment of the climate to emissions is also important in economic models that value climate change, meaning that transient climate response is actually more important to social cost calculations [3]. Additional citations for the ranges of other values in your model would be good. There just seem to be many arbitrary numbers used in the climate model and I can't tell if the citation for MAGICC is supposed to extend a lot further than it seems.

300: Your social discount rates are reasonable but should be justified with reference to literature, e.g. [4, 5].

Figure 12: explain what "MAC sensitivity" is in better words. I don't think the violin plots in the upper section have any meaning, these are robustness checks, not independent observations from one distribution. Similarly, I don't know what the grey bands represent in the lower sections but they don't seem to capture much of the uncertainty. I would also do this plot for 2100 (a year that at policymakers at least sometimes discuss, and has real SSP data) rather than 2450 (a year that isn't modelled well or policy-relevant) and throw this graph in the SI if you really want it.

355: This section is a little confused, since most of it is a summary of what was done, but it claims to be about limitations and future work. Separate out the conclusion from the limitation section.

**SI:**

Figure A2: Why are these not harmonised?

Table A2, etc.: A value of exactly 1,000,000 for a power law relationship is very suspicious and likely indicates either a numerical solution failure or a limit imposed on the relationship not described in the main text. Please investigate and clarify. The fact that $\theta_2$ never goes below 1 but sometimes equals it also seems curious. Why is the number of significant figures so inconsistent between columns?

372: Your cost benefit analysis does not investigate what will happen, but what is conditionally optimal.

**References**

[1] Chris Hope (2013) Critical issues for the calculation of the social cost of $CO_2$: why the estimates from PAGE09 are higher than those from PAGE2002, Climatic Change, https://link.springer.com/article/10.1007/s10584-012-0633-z

[2] Kikstra, J. S., Waidelich, P., Rising, J., Yumashev, D., Hope, C., & Brierley, C. M. (2021). The social cost of carbon dioxide under climate-economy feedbacks and temperature variability. Environmental Research Letters, 16(9), 094037. https://doi.org/10.1088/1748-9326/AC1D0B

[3] Otto, A., Todd, B. J., Bowerman, N., Frame, D. J., & Allen, M. R. (2013). Climate system properties determining the social cost of carbon. *Environmental Research Letters*, *8*(2), 024032. https://doi.org/10.1088/1748-9326/8/2/024032

[4] Greaves, H. (2017). Discounting for public policy: a survey, Econ. Philos., 33(3), 391–439, https://doi.org/10.1017/S0266267117000062.

[5] Nesje, F., Drupp, M. A., Freeman, M. C., & Groom, B. (2023). Philosophers and economists agree on climate policy paths but for different reasons. Nature Climate Change 2023, 1–8. https://doi.org/10.1038/s41558-023-01681-w